# Enhanced Cas12a multi-gene regulation using a CRISPR array separator

Jens P Magnusson[1], Antonio Ray Rios[1], Lingling Wu[1], Lei S Qi[1,2,3]*

[1]Department of Bioengineering, Stanford University, Stanford, United States; [2]Department of Chemical and Systems Biology, Stanford University, Stanford, United States; [3]Stanford ChEM-H Institute, Stanford University, Stanford, United States

**Abstract** The type V-A Cas12a protein can process its CRISPR array, a feature useful for multi-plexed gene editing and regulation. However, CRISPR arrays often exhibit unpredictable performance due to interference between multiple guide RNA (gRNAs). Here, we report that Cas12a array performance is hypersensitive to the GC content of gRNA spacers, as high-GC spacers can impair activity of the downstream gRNA. We analyze naturally occurring CRISPR arrays and observe that natural repeats always contain an AT-rich fragment that separates gRNAs, which we term a *CRISPR separator*. Inspired by this observation, we design short, AT-rich synthetic separators (*synSeparators*) that successfully remove the disruptive effects between gRNAs. We further demonstrate enhanced simultaneous activation of seven endogenous genes in human cells using an array containing the synSeparator. These results elucidate a previously underexplored feature of natural CRISPR arrays and demonstrate how nature-inspired engineering solutions can improve multi-gene control in mammalian cells.

## Introduction

Precise control of cell identity and behavior will require the ability to regulate the expression of many genes simultaneously. Genes can be experimentally turned on or off using transcriptional activators or repressors attached to nuclease-deactivated Cas proteins (dCas) in technologies named CRISPRa and CRISPRi (*Lo and Qi, 2017*). When this Cas fusion protein is recruited to a target gene, guided by a CRISPR-associated guide RNA (*gRNA* for type V CRISPR Cas12, also known as CRISPR-RNAs [crRNAs] when expressed from their native loci) or a chimeric single-guide RNA (*sgRNA* for type II CRISPR Cas9), the target gene can be activated or repressed. While this technology works relatively well for controlling single genes, a central goal is to efficiently control more than a handful of genes at a time for applications in cell engineering.

In their native prokaryotic context, multiple crRNAs are encoded on a CRISPR array transcribed as a single, long transcript (*Brouns et al., 2008*; *Zetsche et al., 2015*; *Figure 1A*). Unlike the widely used Cas9, which requires a trans-activating crRNA (tracrRNA) and RNase III for maturation of its gRNA, Cas12a (also known as Cpf1) can process its own CRISPR array (*Fonfara et al., 2016*; *Zetsche et al., 2015*). Therefore, multi-gene control can be achieved experimentally with Cas12a (*McCarty et al., 2020*). This strategy has recently been used to edit or control the expression of multiple genes in human cells (*Breinig et al., 2019*; *Campa et al., 2019*; *Kleinstiver et al., 2019*; *Tak et al., 2017*; *Zetsche et al., 2017*). However, CRISPR arrays encoding multiple crRNAs often exhibit unpredictable performance for multi-gene regulation. Small-RNA-seq experiments suggest that array processing by Cas12a is uneven, such that the constituent processed gRNAs may differ 10- to 100-fold in abundance (*Campa et al., 2019*; *Liao et al., 2019b*). To reliably regulate many genes experimentally using this system, it will be crucial to optimize the performance of CRISPR arrays in mammalian cells and understand the principles governing CRISPR array processing.

*For correspondence:
stanley.qi@stanford.edu

A gRNA consists of a repeat region, which is often identical for all gRNAs in the array, and a spacer (here used synonymously with 'guide region'), which serves to guide the Cas12a protein to complementary double-stranded DNA (*Figure 1B*). In their natural bacterial setting, the full-length repeat sequence includes a short (~16–18 nt) fragment that gets excised and discarded during gRNA processing and maturation, leaving the final, processed gRNA to consist of a post-processing repeat and a spacer (*Figure 1B*). The excised repeat fragment, which we here denote a *CRISPR separator*, undergoes cleavage at its 3' end by Cas12a itself, and at its 5' end by an unknown bacterial enzyme (*Swarts, 2019*; *Figure 1B*). The separator is not strictly needed for array function and has no known role, as far as we are aware. Cas12a cannot excise the separator on its own, which means that it remains attached to the 3' end of the upstream spacer in mammalian cells (*Zetsche et al., 2017*). For this reason, and because the separator has been seen as dispensable, the separator has been omitted when Cas12a arrays have been experimentally expressed in eukaryotic cells (*Figure 1C*; *Campa et al., 2019*; *Kleinstiver et al., 2019*; *Tak et al., 2017*; *Zetsche et al., 2017*). Yet, despite being under intense evolutionary pressure, the fact that Cas12a arrays from all known bacterial species retain the CRISPR separator suggests that it might have a biological function.

Here, we hypothesized that the separator plays a role in facilitating CRISPR array processing and that array performance is impaired in its absence. We find that gRNAs lacking the separator are highly sensitive to the content of guanine and cytosine (GC) nucleotides in the upstream spacer. Specifically, the higher the GC content of a spacer, the worse the downstream gRNA performs. We notice that the natural separator sequence is rich in adenine and thymine/uracil (AT/U) nucleotides, and we therefore surmised that the separator insulates neighboring gRNAs from one another. We design and insert a synthetic separator (*synSeparator*) consisting of four A/T nucleotides between each gRNA in a Cas12a CRISPR array; this allows more effective CRISPR activation of multiple endogenous genes in human cells. Based on these results, we conclude that the CRISPR separator found in natural Cas12a arrays acts as an insulator that reduces interference between adjacent gRNAs. These results show how natural systems can guide optimization of CRISPR-mediated synthetic multi-gene regulation.

## Results
### gRNA performance is affected by the GC content of the upstream spacer

In bacteria-harboring CRISPR-Cas systems, new spacers are acquired from viral genomes and integrated into the CRISPR array. It is possible that some spacer sequences form RNA secondary structures that interfere with Cas12a processing of the array. RNA secondary structure is in fact known to impede Cas protein binding and processing in similar contexts. For example, Cas12a function is sensitive to hairpin formation downstream of the CRISPR array (*Liao et al., 2019a*), and gRNA processing is impaired by base-pairing of the repeat with regions outside the gRNA (*Creutzburg et al., 2020*). Global RNA structure of the CRISPR array may negatively affect both gRNA processing and performance (*Liao et al., 2019b*). Similarly, the RNA-guided, RNA-cleaving protein Cas13 can also be negatively affected by gRNA secondary structure (*Abudayyeh et al., 2016*; *Yan et al., 2018*). It is therefore theoretically plausible that local secondary structure within the transcribed CRISPR array could interfere with array processing (*Figure 1C*). One feature that promotes RNA secondary structure formation is high GC content (*Chan et al., 2009*) because G-C base pairs contain three hydrogen bonds, compared to A-U's two hydrogen bonds. We first tested whether the GC content of a spacer could affect performance of a downstream gRNA.

To do this, we first developed a method for assembling CRISPR arrays using oligonucleotide hybridization and ligation, with which we could assemble CRISPR arrays containing up to 30 gRNAs (Materials and methods). We used this method to assemble a simple Cas12a array consisting of two consecutive gRNAs whose repeat regions did not contain the separator sequence (*Figure 1D*). In this array, the second gRNA's spacer was complementary to the promoter of GFP, which had been genomically integrated into HEK293T cells (*Figure 1E*). The first gRNA's spacer instead consisted of a non-targeting sequence, which we could alter to study its effect on GFP activation. We refer to this sequence as a dummy spacer.

To study the consequences of varying the dummy spacer, we transfected the HEK293T cells with constructs encoding the CRISPR array and nuclease-deactivated Cas12a (from *Lachnospiraceae*

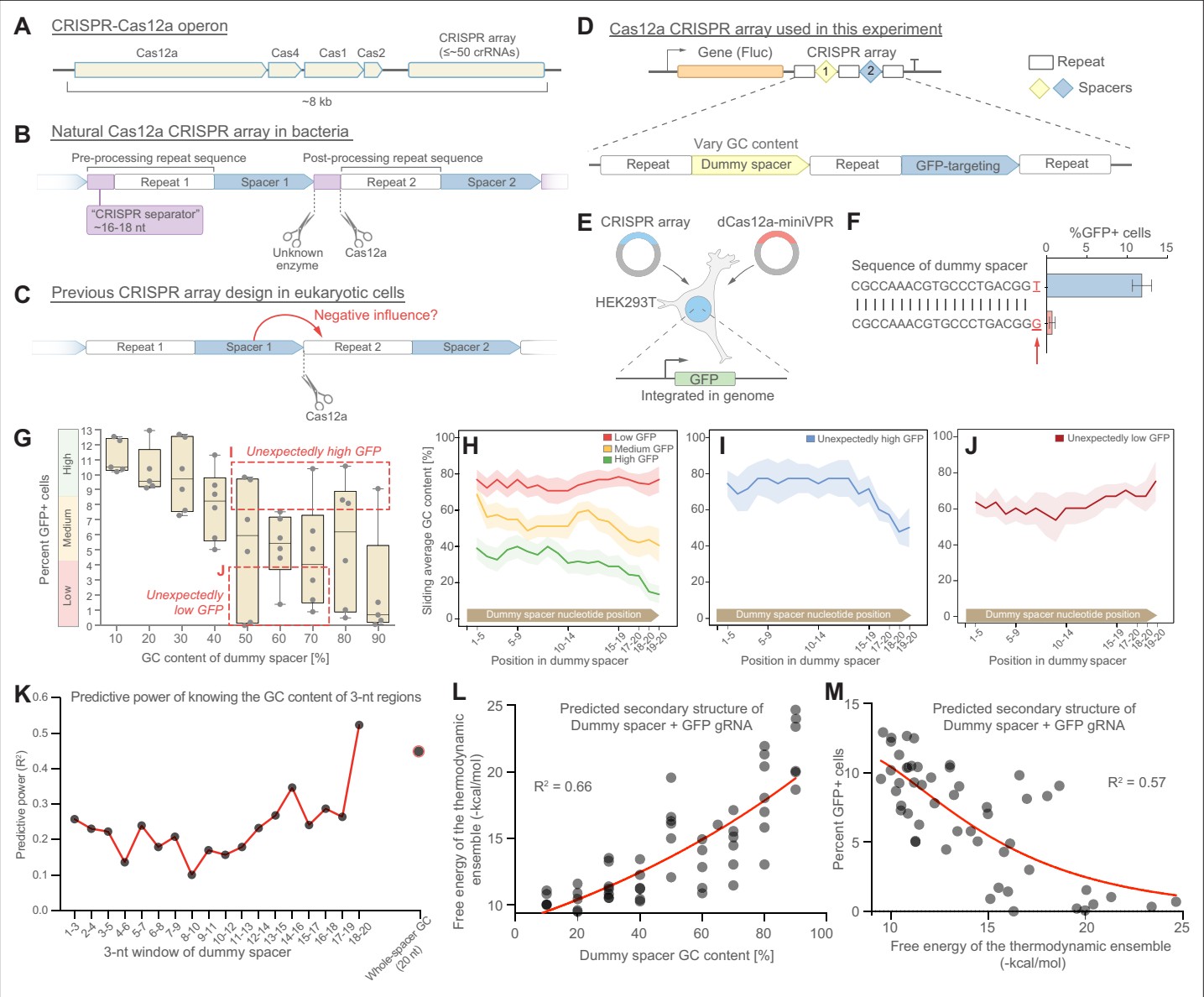

**Figure 1.** gRNA performance is affected by the GC content of the upstream spacer. (**A**) The CRISPR-Cas12a operon consists of Cas genes and a CRISPR array. (**B**) Each gRNA consists of a repeat and a spacer. Pre-processing repeats contain a ~16–18 nt fragment, here denoted *CRISPR separator*, which gets excised by Cas12a and an unknown enzyme. (**C**) The separator has previously been omitted when expressing Cas12a arrays in mammalian cells. We asked if the separator serves to insulate gRNAs from the negative influence of secondary structure in spacers. (**D**) We designed CRISPR arrays consisting of two gRNAs, the first with a non-targeting dummy spacer, and the second targeting the promoter of GFP, genomically integrated in HEK293T cells. (**E**) Experimental setup; Lb-dCas12a-miniVPR was used to activate GFP, and GFP fluorescence was analyzed as a measure of array performance. (**F**) CRISPR arrays can display hypersensitivity to the composition of the dummy spacer. In extreme cases, replacing the last nucleotide from T to G can lead to almost complete abrogation of GFP activation. (**G**) A library of 51 CRISPR arrays, each with a dummy spacer of different GC content. A strong negative correlation is seen between the GC content of the dummy spacer and GFP fluorescence. Each dot represents one of the 51 CRISPR arrays (average of three replicates). Arrays were divided into three groups based on the level of GFP fluorescence they enabled. Boxes indicate two groups that were analyzed in (**I** and **J**). (**H–J**) For each group, the average GC content of a sliding 5-nt window was calculated. The best-performing arrays were the ones where the dummy spacer happened to have low GC content at its 3' end (**H**). Some arrays showed unexpectedly high or low GFP activity for the GC content of their dummy spacers (**G**). These arrays contain low (**I**) or high (**J**) GC content at the very 3' end of their dummy spacers, suggesting that the GC content of the last few bases is an important predictor of array performance. Shaded regions in (**H–J**) represent standard error. (**K**) The predictive power of knowing the GC content of 3-nt regions in the dummy gRNA (Materials and methods). Merely knowing the GC content of the las three bases is more predictive than knowing the overall GC content. (**L**) A plot showing the relationship between GC content of the 51 dummy spacers and the secondary structures they are predicted to form with the GFP-targeting gRNA (the larger the value on the y-axis, the more stable the predicted secondary structure). (**M**) This predicted secondary structure formation is anticorrelated with performance of the GFP-targeting spacer, suggesting that

*Figure 1 continued on next page*

*Figure 1 continued*

strong secondary structures is what impedes array performance.

The online version of this article includes the following figure supplement(s) for figure 1:

**Source data 1.** Raw data used for panel G.

**Source data 2.** Raw data used for panel L.

**Source data 3.** Raw data used for panel M.

**Figure supplement 1.** CRISPR activation of genomically integrated GFP.

**Figure supplement 1—source data 1.** Raw data used for panel C.

---

*bacterium*) fused to the miniaturized VP64-p65-Rta (mini-VPR) tripartite activator and mCherry (subsequently denoted dCas12a-miniVPR) (***Vora et al., 2018***). If the dummy spacer did not influence the performance of the downstream gRNA, we should see equal GFP activation in all experiments. If, on the other hand, the dummy spacer affected the performance of the downstream spacer, we would measure differences in GFP fluorescence. Surprisingly, we found that the CRISPR array sometimes displayed hypersensitivity to the composition of the dummy spacer: In extreme cases, a single nucleotide change from T to G led to almost complete abrogation of GFP activation in transfected cells (***Figure 1F***, Materials and methods).

We asked how such hypersensitivity could come about. We generated a library of 51 CRISPR arrays containing one dummy spacer with varying GC content (10–90%), and one GFP-targeting spacer that was identical in all arrays. Forty-eight hours after transfection, we analyzed cells by flow cytometry and quantified GFP fluorescence as a measure of array performance. Intriguingly, we observed a strong negative correlation between the GC content of the dummy spacer and GFP activation (***Figure 1G***, ***Figure 1—figure supplement 1A***; the percentage of GFP$^+$ cells was a more sensitive measure of array performance than median GFP fluorescence, ***Figure 1—figure supplement 1B***). This indicated that spacers can exert a strong influence on the performance of the downstream gRNA.

Since these dummy spacers were random sequences, we asked whether the distribution of GC content within these spacers mattered for their effect on the downstream gRNA. We first divided all the dummy spacers into three groups based on whether they enabled high, medium, or low GFP activation (***Figure 1G***). For each dummy spacer, we calculated the GC content of a sliding 5-nt window (***Figure 1H***; Materials and methods). Interestingly, this analysis showed that 'permissive' dummy spacers had relatively low GC content at the 3' end, close to the Cas12a cleavage site. In contrast, non-permissive dummy spacers had slightly higher GC content at the 3' end. Dummy spacers that enabled medium-level GFP activation had quite high overall GC content but lower GC content close to the 3' end. This suggested that the GC content of the spacer's last few bases, close to the Cas12a cleavage site, might be a more important determinant than the overall GC content of a spacer.

Surprisingly, dummy spacers in the 50–90% GC range exhibited a wide range of GFP activation, some enabling unexpectedly high GFP activation and others unexpectedly low (***Figure 1G***). We analyzed the sliding GC content specifically of these spacers and found that unexpectedly permissive dummy spacers showed an even stronger trend toward low GC content at the 3' end (***Figure 1I***). In contrast, unexpectedly non-permissive dummy spacers had high GC content at the 3' end (***Figure 1J***). Thus, even if a dummy spacer had high GC content, it could still allow efficient performance of the GFP-targeting gRNA if GC content in the last 3–5 bases was low, and vice versa.

GC content of the entire dummy spacer was moderately predictive of GFP activation ($R^2 = 0.45$; ***Figure 1K***; Materials and methods). However, we found that simply knowing the average GC content of the last three nucleotides in the dummy spacer allowed slightly better predictive power than knowing the GC content of the whole dummy spacer ($R^2 = 0.52$; ***Figure 1K***). The GC content of these last three bases was more predictive of array performance than that of any other three bases in the dummy spacer (***Figure 1K***). These results indicated that high GC content at the 3' end of a spacer impairs performance of the subsequent gRNA.

GC content is a determinant of secondary structure formation. Such secondary structures might interfere with proper folding of the GFP-targeting gRNA. We used the online tool RNAfold (***Lorenz et al., 2011***) to calculate secondary structure formation for the library of 51 dummy spacers with the GFP-targeting gRNA. To quantify the propensity to form secondary structures, we used the calculated free energy of the thermodynamic ensemble, a measure of the stability of the set of structures

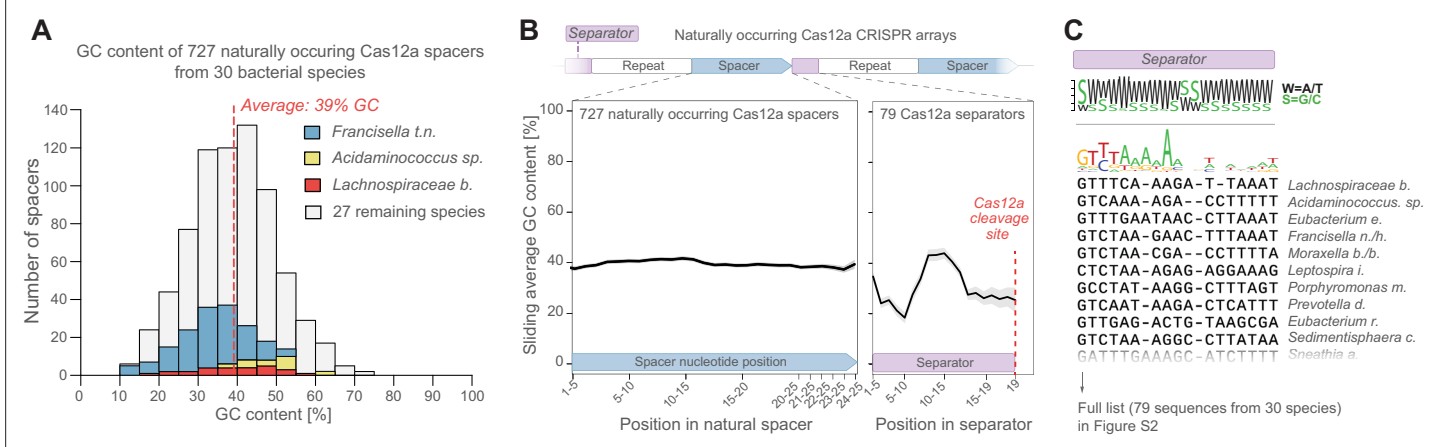

**Figure 2.** Natural CRISPR arrays contain separator sequences with low GC content. (**A**) The GC content of naturally occurring CRISPR-Cas12a spacers display no obvious depletion of high-GC spacers (though spacer GC content is weakly correlated with overall genomic GC content, *Figure 2—figure supplement 1A*). Spacers from commonly used Cas12a variants (Lb, As, Fn) are highlighted in color. (**B**) Neither do these spacers show low GC content at their 3' ends. But the separator sequences of these gRNAs have low GC content. (**C**) This is seen also in a multiple-sequence alignment of 79 natural separator sequences, which suggests that one purpose of the CRISPR separator is to act as an insulator between adjacent gRNAs in a Cas12a CRISPR array.

The online version of this article includes the following figure supplement(s) for figure 2:

**Source data 1.** Raw data used for panel A.

**Figure supplement 1.** Further analysis of naturally occurring crRNA sequences.

adopted by an RNA molecule. We indeed found a strong positive correlation between GC content and predicted secondary structure formation ($R^2$ = 0.66; *Figure 1L*). Accordingly, there was a corresponding anticorrelation between predicted RNA structure and array performance. This anticorrelation was strongest when both the dummy spacer and the subsequent GFP-targeting gRNA were used as input to RNAfold ($R^2$ = 0.57; *Figure 1M*) rather than only the dummy spacer ($R^2$ = 0.27; *Figure 1—figure supplement 1C*). This suggested that high-GC spacers may form secondary structures that are particularly disruptive if they directly involve the downstream gRNA. Taken together, these results suggest that high GC content in spacers can lead to secondary structure formation, which may interfere with proper folding and processing of the CRISPR array transcript.

## Natural CRISPR arrays contain separator sequences with low GC content

We next asked whether bacteria have evolved mechanisms to preferentially incorporate low-GC spacers into their CRISPR arrays to overcome the hypersensitivity to spacer GC content. After all, it could be detrimental to a bacterium if it accidentally incorporated a high-GC spacer that lowered the performance of a pre-existing gRNA. To address this question, we analyzed 727 naturally occurring Cas12a spacer sequences from 30 bacterial species containing the Type V-A CRISPR (*Supplementary file 2*; Materials and methods). However, we did not find any conspicuous absence of GC-rich spacers: Spacer GC content was distributed around an average of 39%, with a range of 10–70% (*Figure 2A*), though spacer GC content did vary between species and was weakly correlated with the overall genomic GC content ($R^2$ = 0.16; *Figure 2—figure supplement 1A*). These results were true also for *L. bacterium*, the species from which our Cas12a variant is derived, and for the commonly used *Acidaminococcus* sp. and *Francisella tularensis* subsp. *novicida* (*Figure 2A*). Neither did we find that GC content was lower at the 3' end of these spacers (*Figure 2B*; Materials and methods). We therefore wondered how natural CRISPR arrays cope with high-GC spacers.

In naturally occurring CRISPR arrays, the separator sequence gets excised through the action of Cas12a and an unknown enzyme (*Figure 1B*; *Zetsche et al., 2015*). We asked whether the separator might act as an insulator that protects every gRNA from disturbances caused by the secondary structure in upstream spacers. We analyzed 79 unique separator sequences from 30 bacterial species (*Supplementary file 2*; Materials and methods). Overall, these separators showed little sequence

conservation (*Figure 2C*, *Figure 2—figure supplement 1B*), except for a moderately conserved region at the very 5' end (GTYTA). This conserved region possibly acts as a recognition motif for the unknown enzyme responsible for cleaving the separator at the 5' end. However, we did detect a strong bias for low GC content in the 79 separator sequences (*Figure 2B–C*). This opened the possibility that in natural CRISPR arrays, the separator aids CRISPR array processing by providing an AT-rich sequence that maximizes Cas12a accessibility to its cleavage site.

Like Cas12a, the type VI CRISPR RNA-cleaving enzyme Cas13d can also process its own gRNA (*Konermann et al., 2018*; *Yan et al., 2018*; *Zhang et al., 2018*; *Zhang et al., 2019*), an ability that makes Cas13d attractive for multi-gene regulation on the RNA level (*Figure 2—figure supplement 1C*). We similarly analyzed the 6-nt CRISPR separator from 12 bacterial species containing Cas13d arrays (*Supplementary file 2*; Materials and methods). We found a strong bias for low GC content here as well, particularly at the 3' end of the separator, close to the Cas13d cleavage site (*Figure 2—figure supplement 1C*). This suggests that sensitivity to spacer GC content is a factor that has shaped the evolution of multiple CRISPR-Cas systems.

## A synthetic separator improves CRISPR array performance in human cells

We asked whether Cas12a CRISPR arrays would show improved performance in human cells if they included the natural CRISPR separator between each gRNA. To test this, we designed CRISPR arrays where the first gRNA contained a dummy spacer (30 % GC content) and the second a GFP-targeting spacer, with and without the natural, 16-nt separator from *L. bacterium* preceding each repeat (*Figure 3A*; Materials and methods). Including this separator, however, almost completely abolished the array's function, as almost no GFP activation was seen in the transfected cells (*Figure 3B*). This is consistent with a previous study that reported poor performance of gRNAs containing the pre-processed Cas12a repeat downstream of the gRNA of interest, which contains the separator (*Liu et al., 2019*), and is why previous designs have completely excluded any separator sequence. One possible reason is that the long separator remains attached at the 3' end of each spacer because Cas12a cannot fully excise it (*Zetsche et al., 2015*; *Zetsche et al., 2017*), which might interfere with Cas12a function, as has been shown previously (*Nguyen et al., 2020*).

We instead asked if we could incorporate only a portion of the separator and still retain its insulating function. We generated CRISPR arrays in which all gRNAs were preceded either by 1–4 A/T nucleotides from the natural *L. bacterium* separator or by a single G as a control (*Figure 3A,C*). We generated three versions of each array, where the GC content of the dummy spacer was 3%, 50 %, or 70 %. (We used the 50% and 70% spacers that had resulted in the lowest level of GFP activation in *Figure 1G*; *Supplementary file 3*.) Interestingly, addition of an AT-rich synthetic separator improved the performance of the CRISPR array in all cases (*Figure 3D*), suggesting that a very short, AT-rich synthetic separator was sufficient to counteract the interference between gRNAs. We denoted this new sequence a *synSeparator* for synthetic separator.

Guide-RNA performance can be increased by adding RNA extensions to the 3' end of the gRNA's spacer (*Creutzburg et al., 2020*; *Kocak et al., 2019*; *Nguyen et al., 2020*). We asked whether the increased GFP activation we observed with the AAAT synSeparator was attributable to the AAAT extension added to the 3' end of the GFP-targeting spacer, rather than by the AAAT separating the two gRNAs in this array. We made four variants of this array where we changed the position of the G or AAAT separators (*Figure 3E*). We observed activated GFP expression with *variant 2* but not with *variant 3* (*Figure 3F*), supporting the hypothesis that the synSeparator exerts its main effect by facilitating gRNA insulation and processing. We did observe a small additional increase in effectiveness using *variant 4* compared to *variant 2*. This can either be attributed to improved insulation and processing at the 3' end of the GFP spacer itself (which has 45 % GC content) or to increased performance of the GFP-targeting spacer due to the 3' AAAT extension on this spacer.

We surmised that computational prediction of RNA structure might yield insights into the mechanism by which the CRISPR separator facilitates gRNA processing. We used RNAfold to analyze two of the worst-performing dummy spacers from *Figure 1G* (50% and 70% GC) (*Figure 3G*). This analysis suggested that these spacers form secondary structures that involve the GFP-targeting gRNA and additionally restrict Cas12a's access to its cleavage site (*Figure 3H*). Interestingly, addition of the AT-rich natural separator, or the synSeparator, appeared to loosen up these structures or create

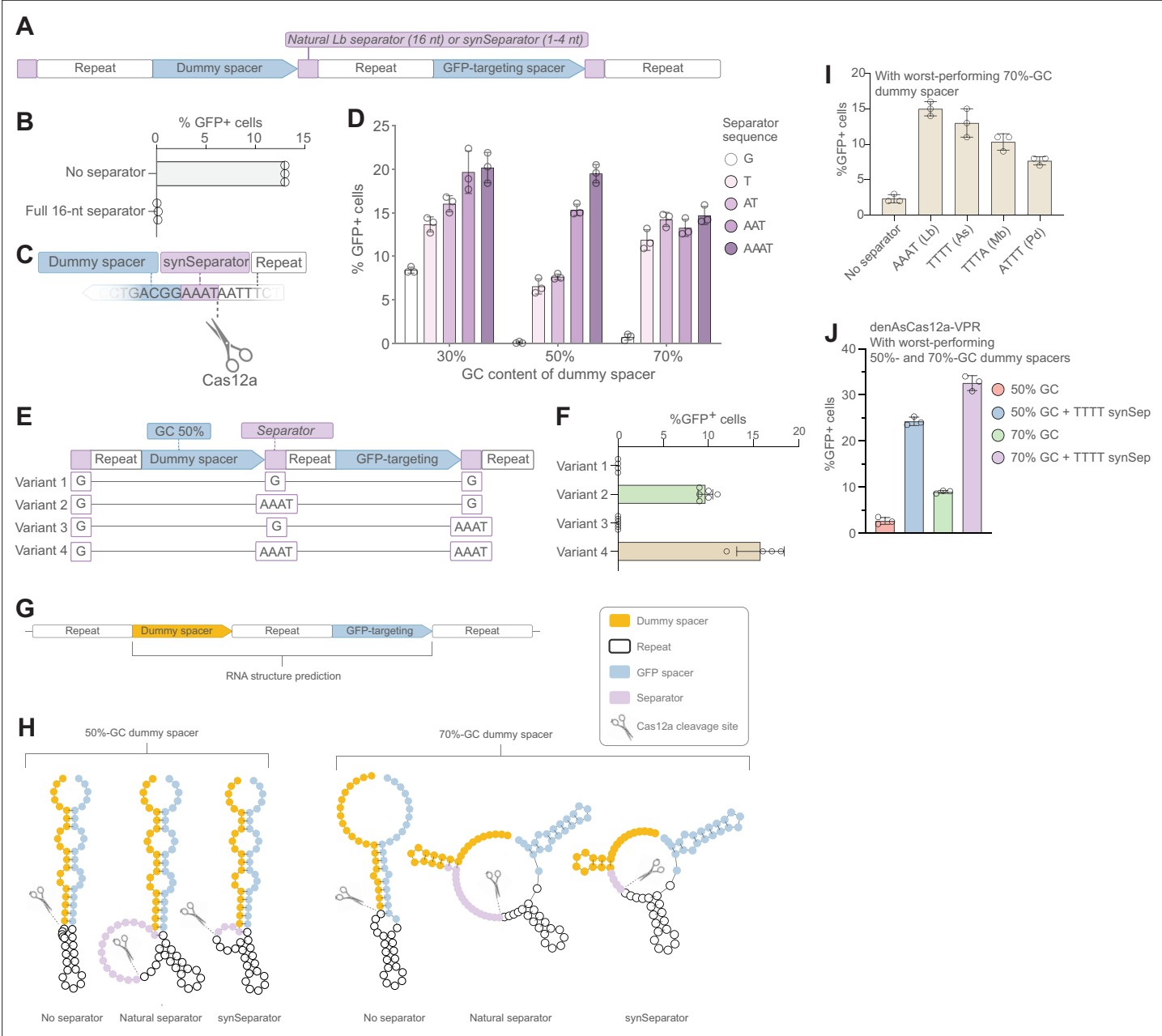

**Figure 3.** A synthetic separator improves CRISPR array performance in human cells. (**A**) A diagram showing how we introduced either the full, 16-nt separator sequence from *L. bacterium* or a short 1–4 nt synthetic separator (synSeparator) between each gRNA in the CRISPR array. (**B**) The full, 16-nt separator sequence almost completely eliminates GFP activation. (**C**) Diagram showing inclusion of synSeparator sequences from the 3′ end of the *L. bacterium* in the CRISPR array. Each array contained a synthetic separator (G, T, AT, AAT, or AAAT) upstream of every repeat. The GC content of the dummy spacer was 30%, 50 %, or 70 %. The most non-permissive 50%- and 70%-GC spacers from ***Figure 1G*** were used. (**D**) AT-rich synSeparators improve GFP activation compared with a single G nucleotide. (**E, F**) This effect is caused by the separator upstream of the GFP-targeting gRNA: Adding the synSeparator only downstream of the GFP-targeting gRNA leads to no improvement. This suggests that the separator acts by improving gRNA processing rather than by generating a 3′ AAAT overhang on the GFP-targeting spacer itself. (**G**) RNA secondary structure prediction of two of the most disruptive dummy spacers (50% and 70% GC) together with the subsequent GFP-targeting gRNA. (**H**) This suggests that these spacers form secondary structures that involve the GFP-targeting spacer. Both the natural *L. bacterium* separator and the AAAT synSeparator are predicted to break up these structures or form protrusions that may facilitate Cas12a access to its cleavage site. (**I**) SynSeparators derived from other bacterial species (see ***Figure 2C***) can rescue poor GFP activation caused by a non-permissive dummy spacer (70 % GC) in a CRISPR array (array design as in **A**). (**J**) The enhanced Cas12a protein from *A. species* (***Kleinstiver et al., 2019***) is also sensitive to GC content of a dummy spacer (array design as in **A**) and its performance can be rescued using a TTTT synSeparator derived from its natural separator.

The online version of this article includes the following figure supplement(s) for figure 3:

*Figure 3 continued on next page*

*Figure 3 continued*

**Source data 1.** Raw data used for panel D.

**Source data 2.** Raw data used for panel F.

**Figure supplement 1.** Targeted design of disruptive secondary structure, and no positional effect of crRNAs within a CRISPR array.

an accessible bulge at the Cas12a cleavage site (*Figure 3H*). This suggested that, for spacers prone to forming obstructive secondary structures, AT-rich separators give Cas12a improved access to its cleavage site.

We asked whether the GFP-targeting gRNA would be most sensitive to secondary structures that form exclusively within the dummy spacer or to structures that also involve the repeat region of the GFP-targeting gRNA. To address this question, we generated several variants of a short CRISPR array guided by RNAfold secondary structure prediction, where we gradually mutated the dummy spacer to form secondary structures that were either limited to the dummy spacer or that competed with the natural pseudoknot structure in the GFP-gRNA's repeat region (*Figure 3—figure supplement 1A*, *Supplementary file 1*). Results showed that, as the predicted extent of base-pairing increased, CRISPRa performance gradually worsened (*Figure 3—figure supplement 1B*). Interestingly, this happened both when the secondary structure was limited to the dummy spacer and when it involved the GFP-targeting gRNA's repeat region. This indicated that CRISPR array performance is sensitive to both kinds of disruption. We then included the AAAT synSeparator to these worst-performing arrays and found that this rescued CRISPRa performance in both cases (*Figure 3—figure supplement 1B*), suggesting that the separator acts both by physically insulating gRNAs from each other and by breaking up secondary structures that involve two consecutive gRNAs.

Although natural CRISPR-separators are all AT-rich, they show little sequence conservation. We asked whether synSeparators derived from other species than *L. bacterium* would be equally effective at rescuing poor-performing arrays. We generated short CRISPR arrays containing one of the worst-performing dummy spacers (70 % GC; *Figure 3A*) and included 4-nt synSeparators derived from the 3' ends of other natural separators (*Figure 3I*; compare *Figure 2C*). Indeed, all these synSeparators improved performance of the GFP-targeting gRNA (*Figure 3I*), indicating that the effect was not sequence specific to the AAAT synSeparator.

Next, we asked whether Cas12a variants from other species than *L. bacterium* are also sensitive to spacer GC content. We used the previously engineered nuclease-deactivated, enhanced Cas12a from *A. species* (denAsCas12a) (*Kleinstiver et al., 2019*) and designed short CRISPR arrays (*Figure 3A*) containing two of the worst-performing dummy spacers (50% and 70% GC), the natural, post-processing repeat region of *A. species*, and a synSeparator derived from the natural *A. species* separator (TTTT) (*Supplementary file 2*). Indeed, denAsCas12a was also sensitive to spacer GC content, and its performance was improved by the TTTT synSeparator (*Figure 3J*), indicating that sensitivity to spacer GC content is not limited to LbCas12a.

It is conceivable that a CRISPR array transcript can assume global secondary structures, in which case a gRNA's position within the array might affect its performance depending on the structure of surrounding regions. Indeed, such a positional effect has been observed in *E. coli* (*Liao et al., 2019b*). To test whether we would observe such a positional effect, we designed four CRISPR arrays consisting of one GFP-targeting spacer and three dummy spacers (30–33% GC; *Supplementary file 4*), where the GFP-targeting spacer was in a different position in each array (*Figure 3—figure supplement 1C*). We made sure that the GFP-targeting gRNA would be preceded by the same dummy gRNA in as many arrays as possible, as different dummy gRNAs may have different propensities to generate secondary structures immediately upstream of the GFP-targeting gRNA. The synSeparator was not included for this experiment. We observed no difference in GFP activation between any of the four arrays (*Figure 3—figure supplement 1D*). This indicates that position within a CRISPR array is not necessarily an important determinant of gRNA performance but may become important when the CRISPR array assumes tight secondary structures within a specific region as observed in Liao et al.

## The synSeparator improves CRISPR array processing

We hypothesized that the synSeparator improves array performance by facilitating Cas12a processing and performed an in vitro cleavage assay to test this hypothesis. Using in vitro transcription, we

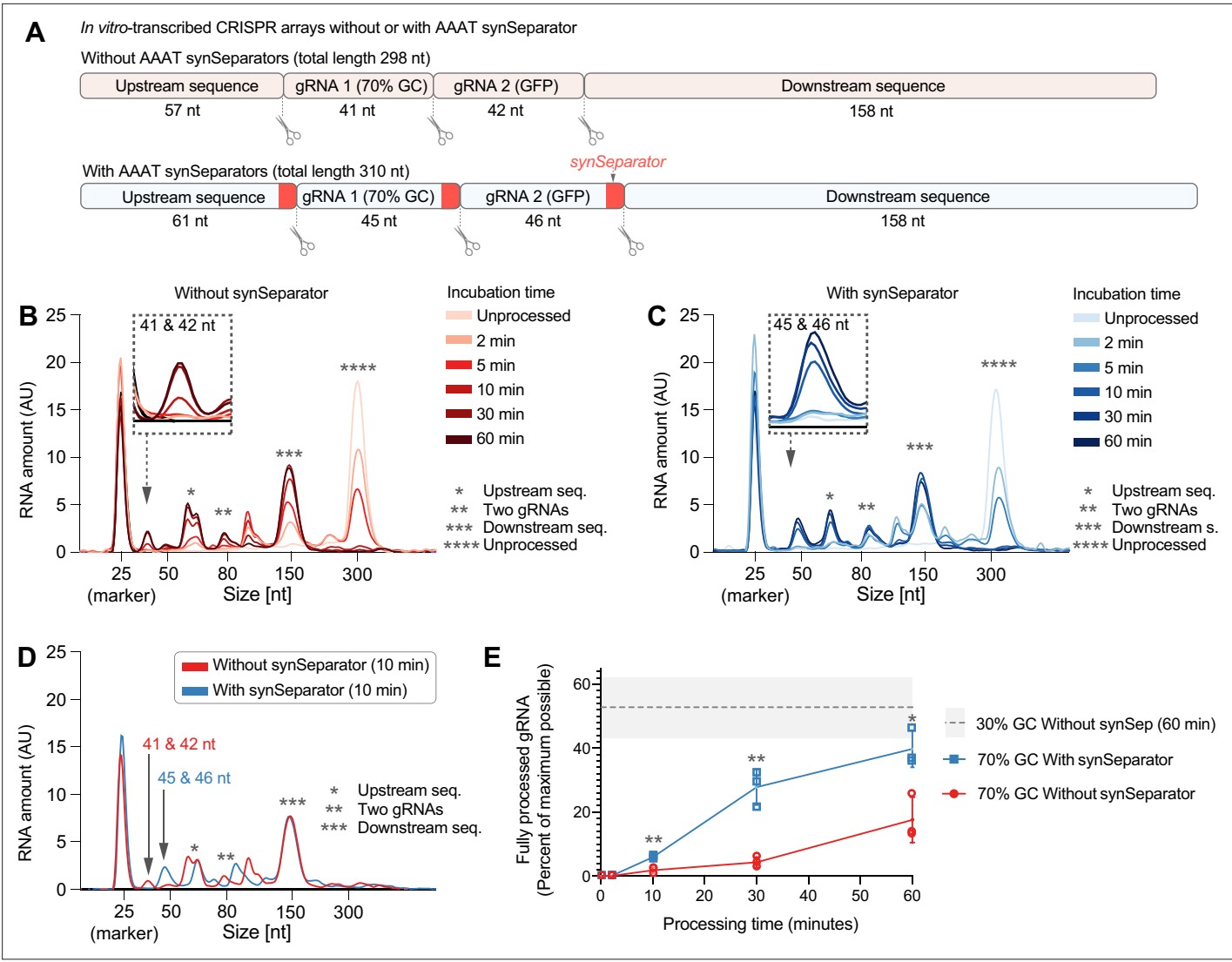

**Figure 4.** The synSeparator improves CRISPR array processing. (**A**) In vitro-transcribed CRISPR arrays containing one of the worst-performing dummy spacers (70 % GC), either with or without the AAAT synSeparator upstream of each Cas12a cleavage site. (**B, C**) Representative Bioanalyzer electropherograms show how both CRISPR arrays become more fully processed with longer Cas12a incubation times (Asterisks highlight specific cleavage products. Note the 25-nt RNA marker, which is not part of the CRISPR array, and how its peak height varies due to sample normalization. See *Figure 4—figure supplement 1* for separate plots). However, the synSeparator-containing array is processed more efficiently, seen clearly after a 10 min incubation (**D**; note that the synSeparator remains attached to the gRNA and slightly increases its length). Bioanalyzer peak data were used to calculate how much of the maximum possible processing had occurred at different time points (**E**; calculations presented in Materials and methods; n = 3 replicates for each time point). For reference, a gray dotted line shows how much processing had occurred in a CRISPR array containing a 30%-GC dummy spacer and no synSeparator after a 60 min incubation (n = 5 replicates). AU, arbitrary units. Error bars and the gray area in (**E**) indicate standard deviation.

The online version of this article includes the following figure supplement(s) for figure 4:

**Source data 1.** Raw data used for panel A.

**Figure supplement 1.** Individual Bioanalyzer electropherograms of Cas12a cleavage time series.

generated RNA transcripts of one of the worst-performing two-gRNA arrays containing a 70 % GC dummy spacer (see *Figure 3A,D*), either with or without the AAAT synSeparator upstream of each Cas12a cleavage site (*Figure 4A*, *Supplementary file 4*). We incubated this array with Lb-Cas12a protein for different amounts of time (2, 5, 10, 30, and 60 min) and analyzed the cleavage products on a Bioanalyzer (Materials and methods). The resulting electropherograms (*Figure 4B–D*) were normalized based on RNA amount to make the results quantitatively comparable across biological replicates.

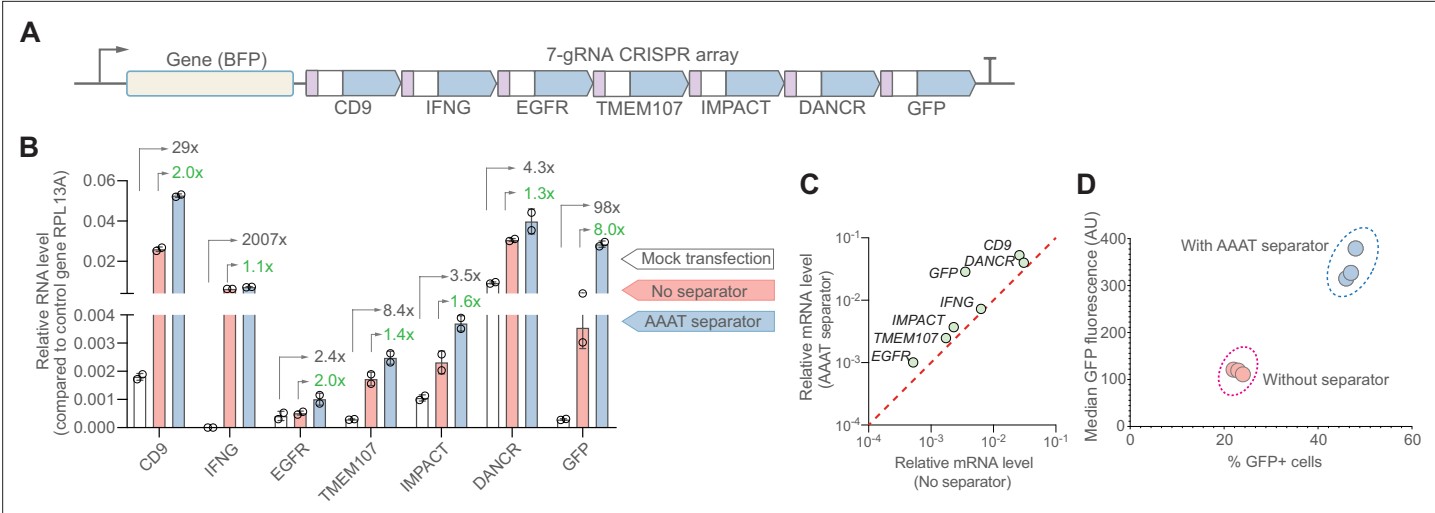

**Figure 5.** The synthetic separator improves multiplexed gene activation in human cells. (**A**) We designed a 7-gRNA array to activate seven endogenous/reporter genes in HEK293T cells with and without the AAAT synSeparator between each gRNA. (**B**) For all target genes, the synSeparator improves target gene activation level, as measured by RT-qPCR. Arrows indicate the expression fold-change achieved with the synSeparator compared to mock transfection (black values) or the CRISPR array not containing the synSeparator (green values). (**C**) Plot showing that the improvement of CRISPR activation is consistent across genes. (**D**) This improvement is also seen on the protein level for the target gene GFP, as measured by GFP fluorescence and percent GFP-positive cells.

The online version of this article includes the following figure supplement(s) for figure 5:

**Source data 1.** Raw data used for panel B.

The results showed that array processing occurred both in the presence and absence of the synSeparator but that it was more efficient with the synSeparator. As Cas12a incubation proceeded for longer times, more single gRNAs were excised (41/42 nt and 45/46 nt in length without and with the AAAT synSeparator, respectively; *Figure 4B,C*; see *Figure 4—figure supplement 1* for each point plotted separately), but at all time points more single gRNAs had been processed from the array containing the synSeparator (*Figure 4E*; see Materials and methods for calculations). Though the exact numbers may not correspond to what would be seen inside living cells, these results indicate that the synSeparator improves Cas12a processing efficiency of CRISPR arrays.

## The synthetic separator improves multiplexed gene activation in human cells

We next tested if the addition of the synSeparator would improve CRISPR activation of endogenous genes when gRNAs were expressed in a CRISPR array. We transfected HEK293T cells (*Figure 1E*) with one of two CRISPR arrays containing seven gRNAs (*Figure 5A*), one array with and one array without the AAAT synSeparator between each gRNA. Each gRNA targeted the promoter of a different gene for activation. We co-transfected a dCas12a activator containing full-length VPR (*Chavez et al., 2015*). Target genes included both protein-coding genes (*CD9, IFNG, EGFR, TMEM107, GFP*) and long non-coding RNAs (*IMPACT, DANCR*) (*Supplementary file 3*; Materials and methods). The seven target genes differed in their baseline expression levels in HEK293T cells (*Figure 5B*). We analyzed target gene levels using RT-qPCR on bulk-RNA isolates and did not sort transfected cells based on uptake of the Cas12a or CRISPR array plasmids.

Including the synSeparator increased activation levels of all target genes compared to the array without the synSeparator (*Figure 5B*). The improvement effect varied, ranging from 1.1-fold to 8.0-fold, but was consistent across all target genes (*Figure 5C*). The increase was also seen on the protein level, which we could analyze for GFP via flow cytometry (*Figure 5D*). These results indicate that addition of a short, AT-rich sequence between gRNAs in a Cas12a CRISPR array can improve multi-gene activation in HEK293T cells.

## Discussion

In this study, we found that Cas12a CRISPR arrays are sensitive to spacer GC content (*Figure 1G*). Specifically, high GC content at the 3' end of a spacer leads to poor performance of the downstream gRNA (*Figure 1H–K*). This is likely caused by secondary structures (*Figure 1M*) that are either restricted to a single gRNA or involve two consecutive gRNAs (*Figure 3—figure supplement 1A,B*). CRISPR arrays containing such disruptive gRNAs are processed inefficiently by Cas12a (*Figure 4B–E*), which likely explains their poor performance. However, AT-rich separator sequences, even short, 4-nt sequences, can break up such secondary structures and give Cas12a access to its cleavage site (*Figure 3H*), improving CRISPR array performance (*Figure 3D–J*). Consequently, including short synSeparators between the gRNAs in a Cas12a array is a simple way to improve CRISPRa performance in human cells (*Figure 5*).

### Considerations for activation of endogenous genes

We note that the improvement afforded by the synSeparator was not as large for the endogenous genes as for our experimental system with the worst-performing dummy spacers. We speculate that this may be for either of two reasons. First, for the CRISPR arrays in *Figure 5*, we used spacers that were either derived from previously published studies or had been pre-tested in our lab. All these spacers likely represent the best of several tested spacers and had thus already been shown to work even in the absence of a synSeparator. We did not necessarily select them for being GC-rich or prone to secondary structure formation. Therefore, researchers may experience the greatest benefit from the synSeparator with spacers that have not already proved functional without the synSeparator. Second, our clonal HEK293T cell line likely carries several copies of the GFP gene in its genome, each of which has seven protospacers in its TRE3G promoter. This multitude of Cas12a binding sites may serve to amplify the CRISPRa effect compared to endogenous genes, which only have two Cas12a binding sites, one in each copy of the gene (though the aneuploid HEK293T cells may have more).

One limitation of our study was that we only investigated the performance of the synSeparator in HEK293T cells and not other cell types. HEK293T cells are easy to transfect and readily express a large protein like dCas12a-VPR-mCherry (>6 kb). We nevertheless hypothesize that the negative influence of high-GC spacers and the insulating effect of synSeparators are generalizable across cell types. That is because we could observe these effects even in the cell-free context of an in vitro expression system (*Figure 4*). This suggests that the sensitivity to spacer GC content is determined only by the interaction between Cas12a and the array, rather than being dependent on a particular cellular context. We surmise that the synSeparator will be most beneficial in cells that can readily express a dCas12a activator and preferentially for high-GC spacers that have not already been shown to work even without a synSeparator.

### GFP as readout of CRISPR array performance

We used an experimental system where GFP fluorescence served as a readout of CRISPR array performance. In this system, the *percentage of GFP+ cells* was a more sensitive measure of array performance than *median GFP fluorescence* (*Figure 1G*, *Figure 1—figure supplement 1*). Even low levels of *GFP* transcription leads to bright GFP fluorescence. We used a relatively weak Cas12a activator (Lb-d-Cas12a driven by the PGK promoter, with mini-VPR rather than full-length VPR), where the percentage of GFP+ cells rarely exceeded 20. This was deliberate; the combination of a weak Cas activator and a GFP-percentage readout was ideal for measuring small differences in CRISPR array performance. We speculate that in cases where a very strong Cas activator is used (where most cells activate GFP), median GFP fluorescence may be a better measure.

### Suggested mechanism of action for the synSeparator

In our experiments, including the full 16-nt separator from *L. bacterium* in a CRISPR array led to an almost complete failure to activate the target gene (*Figure 3B*). At face value, this would seem to contrast with previous studies that reported efficacious CRISPR editing even when including the separator (*Liu et al., 2019*; *Zetsche et al., 2017*). But we think these data actually support our findings. First, in Liu et al., the pre-processed repeat (including the separator) shows improved performance if it is positioned upstream of the gRNA. If one such repeat was additionally placed downstream of the gRNA, this construct performed the worst of all tested constructs. This is

consistent with our conclusion that the full-length separator reduces performance of gRNAs when present in CRISPR arrays in mammalian cells. Second, our array in *Figure 3B* is not entirely non-functional. Re-running the same experiment with a stronger Cas12a-activator (Cas12a-VPR; also used in *Figure 5*) shows some GFP activation even with the full separator (1.4% vs 20.8 % GFP$^+$ cells). But for consistency, we have used the same, slightly less effective, Cas12a activator (dCas12a-miniVPR) for all GFP-targeting experiments. Third, both the Liu et al. and Zetsche et al. studies used CRISPR editing rather than CRISPRa. We speculate that this might explain the relatively high indel frequency: Only a single cleavage event needs to take place for an indel to occur, whereas gene activation presumably requires the Cas12a activator to be present on the promoter for extended periods of time.

We found that the AAAT synSeparator improves the ability of Cas12a to process a poor-performing CRISPR array containing a 70%-GC dummy spacer (*Figure 4*). This result supports our observations that spacer GC content is correlated with predicted secondary structure formation (*Figure 1L*) and that GC content close to the Cas12a cleavage site is particularly predictive of the performance of the downstream gRNA (*Figure 1K*). Furthermore, previous data show that Cas12a processing is sensitive to RNA secondary structure (*Creutzburg et al., 2020*). Thus, it is likely that the synSeparator improves performance of CRISPR arrays in vivo by increasing Cas12a processing efficiency. In our Cas12a cleavage assay, we observed that array processing still occurred even without the synSeparator, though it was less efficient than with the synSeparator. These experimental conditions do not necessarily reflect the situation inside living cells. For example, we used an excess of Cas12a protein (16:1 Cas12a:array ratio), which may not be the case inside cells. Furthermore, our in vitro conditions in principle allow the reaction to proceed to completion (even though the reaction appears to slow down after 30 min, possibly due to the lower concentration of available Cas12a molecules as more and more Cas12a molecules become sequestered by their RNA cleavage products and unavailable to participate in further processing events). This may not be true in a living eukaryotic cell, where RNA turnover may constantly remove partially processed CRISPR arrays. Under these conditions, slow processing kinetics may serve to prevent the complete processing of gRNAs.

One additional possible mechanism for how the synSeparator improves CRISPR array performance could be that the synSeparator improves the GFP-targeting spacer itself rather than facilitating gRNA processing. In other words, that it is the synSeparator downstream of the GFP-targeting spacer, rather than the one upstream, that improves performance by remaining attached to the post-processed GFP-targeting spacer. This would be in line with studies showing that gRNA performance can be improved by RNA 3′ spacer extensions (*Creutzburg et al., 2020*; *Kocak et al., 2019*; *Nguyen et al., 2020*). However, our results demonstrate that the synSeparator exerts its beneficial effect before or during gRNA processing, rather than by modifying the 3′ end of the post-processed GFP-targeting spacer itself (*Figure 3F*; *version* 2 vs. *version* 3; *Figure 4*). In addition, RNA secondary structure prediction (RNAfold) did not indicate that the GFP-targeting spacer would fold back on itself when an AAAT extension is added to the 3′ end, which would have been the case for the mechanism demonstrated by Creutzburg et al. (data not shown). Although we cannot exclude that a 3′ extension to the GFP-targeting gRNA itself contributes to gRNA efficacy (*Figure 3F*, *version* 3 vs. *version* 4), such an effect would be marginal.

A competing explanation for the synSeparator's function could be that the natural and synthetic separators provide a specific sequence motif that aids Cas12a binding and/or processing rather than counteracting secondary structure formation. However, natural separators contain no conserved sequence close to Cas12a's binding site that could act as such a binding motif (*Figure 2C*). And we found that different synSeparators can rescue poor array performance (*Figure 3I*). Thus, this possibility is unlikely.

It is conceivable that our introduction of AAAT synSeparators could introduce polyadenylation sites (e.g., AATAAA, ATTAAA, AGTAAA, etc.) (*Beaudoing et al., 2000*), which could prematurely terminate the transcript and reduce CRISPRa activity. However, we analyzed all sequences and found that this was not the case. In any case, polyadenylation sites are AT-rich, so if this was an issue, AT-rich dummy spacers would be expected to perform worse than GC-rich ones, which is the opposite of what was observed (*Figure 1G*).

## Generalizability and outlook

We found that the beneficial effect of adding an AT-rich synSeparator was not limited to the AAAT sequence derived from the natural separator of *L. bacterium*; three other AT-rich synSeparators also rescued the detrimental effects of a high-GC upstream spacer, though with various efficacy (*Figure 3I*). We speculate that the quantitative difference between the synSeparators could either be due to intrinsic insulation capacity of these sequences or the way they interact with Lb-Cas12a, or sequence-specific interactions with this particular CRISPR array.

It is informative to compare the separator of Cas12a with that of Cas13d. Cas13d's pre-processed repeat also contains an AT-rich separator sequence. Like Cas12a, Cas13d itself can only cleave the 3′ end of its separator (*Yan et al., 2018*). But Cas13d gRNAs nonetheless perform well in mammalian cells despite the separator remaining attached to the spacer. Perhaps this is because the Cas13d separator is only six nt long, in contrast to Cas12a's 16–18 nt. A recent study found that the performance of Cas12a gRNAs starts to drop if RNA sequences > 13 nt long are attached to the 3′ end of the spacer (*Nguyen et al., 2020*), consistent with this hypothesis.

As a biotechnology, improvement of simultaneous gene targeting is beneficial to several applications. In this work, we demonstrated improvement of endogenous gene activation using the redesigned CRISPR array containing the synthetic separator, but it is possible that the same design can be used for multiplexed gene editing and base editing using the Cas12a system. Furthermore, because CRISPR-mediated multi-gene activation has been increasingly used for stem cell reprogramming and large-scale genetic screening (*Replogle et al., 2020*; *Weltner et al., 2018*), our method will further enhance these emerging applications. Taken together, the results from our study suggest that the CRISPR separator has evolved as an integral part of the repeat region and that one of its functions is to insulate gRNAs from the disrupting effects of varying GC content in upstream spacers. This insulating effect of AT-rich CRISPR separators may be important enough to have helped shape the evolution of Cas12a CRISPR arrays. Furthermore, our study demonstrates how design inspired by the natural CRISPR system can improve the efficacy of tools for CRISPR gene editing and regulation.

## Materials and methods

### Key resources table

| Reagent type (species) or resource | Designation | Source or reference | Identifiers | Additional information |
|---|---|---|---|---|
| Cell line (*Homo sapiens*) | HEK Lenti-X 293T (modified) | Takara Bio | 632,180 | Genomically integrated TRE3G-GFP |
| Commercial assay or kit | iScript cDNA Synthesis Kit | Bio-Rad | 1708890 | |
| Commercial assay or kit | RNeasey Plus Mini Kit | Qiagen | 74,134 | |
| Commercial assay or kit | iTaq Universal SYBR Green Supermix | Bio-Rad | 1725120 | |
| Peptide, recombinant protein | T7 DNA Ligase | NEB | M0318S | |
| Peptide, recombinant protein | T4 Polynucleotide Kinase | NEB | M0201L | |
| Commercial assay or kit | NucleoSpin Gel and PCR Clean-up | Takara | 32–740609.50 | |
| Commercial assay or kit | In-Fusion HD Cloning Plus | Takara | 638,920 | |
| Commercial assay or kit | KAPA HiFi Hot-start PCR kit | Roche | 07958897001 | |
| Commercial assay or kit | HiScribe T7 Quick High Yield RNA Synthesis Kit | NEB | E2040S | |
| Commercial assay or kit | MEGAclear Transcription Clean-Up Kit | Thermo Fisher | AM1908 | |
| Peptide, recombinant protein | EnGen Lba Cas12a (Cpf1) - 2000 pmol | NEB | M0653T | |
| Peptide, recombinant protein | Recombinant RNase inhibitor | Takara Bio | 2313 A | |
| Commercial assay or kit | RNA Nano 6,000 chips | Agilent | 5067–1511 | |
| Sequence-based reagent | Low Range ssRNA Ladder | NEB | N0364S | |
| Commercial assay or kit | TransIT-LT1 transfection reagent | Mirus Bio | MIR2304-70 | |

## Cell lines

We used HEK293T cells (Takara Bio, Japan). The identity and negative mycoplasma status of these cells have been confirmed by the vendors. Upon receiving the cell line, we engineered the cells to carry a genomically integrated dscGFP gene driven by the TRE3G promoter (consisting of seven repeats of the Tet response element) (*Kempton et al., 2020*). This cell line was clonally sorted and expanded and showed no background GFP fluorescence. We expanded and kept multiple aliquots in the liquid nitrogen freezer. The cells were taken out of the liquid nitrogen freezer for experiments and discarded after maximum 3 months of use.

## Cell culture

Cells were cultured in DMEM+ GlutaMAX (Thermo Fisher, Waltham, MA) containing 100 U/ml of penicillin and streptomycin (Thermo Fisher) and 10 % fetal bovine serum (Clontech). Cells were grown at 37 °C with 5 % $CO_2$ and passaged using 0.05 % Trypsin-EDTA solution (Thermo Fisher) or TryplE Express Enzyme (Thermo Fisher).

## Transfection

Cells were transfected with constructs carrying (1) the nuclease-deactivated (D832A) dCas12a (from *L. bacterium*, human codon-optimized) (*Zetsche et al., 2015*) fused either to the VP64-p65-Rta (VPR) activator (*Chavez et al., 2015*) and mCherry (*Figure 5*), or to mini-VPR (*Vora et al., 2018*) and mCherry (remaining experiments); (2) a CRISPR array-expressing plasmid. For *Figures 1 and 3a*, CRISPR array construct consisting of firefly luciferase immediately followed by a CRISPR array and an SV40 pA terminator, expressed under the CAG promoter element, was used (*Supplementary file 1*). For the activation of seven endogenous genes (*Figure 5*), firefly luciferase was replaced with BFP and a Malat1 Triplex sequence *Campa et al., 2019* followed by the *L. bacterium* Cas12a leader sequence (*Supplementary file 1*).

Cells were seeded 1 day before transfection at a density of $5 \times 10^4$ cells per well in a 24-well plate. Cells were transfected using TransIT-LT1 transfection reagent (Mirus Bio, Madison, WI) according to the manufacturer's recommendation (250 ng dCas12a-VPR-mCherry plasmid; 250 ng CRISPR array plasmid; 1.5 µl transfection reagent per well).

## Flow cytometry

Two days after transfection, cells were dissociated using 0.05 % Trypsin-EDTA or TrypLE (Thermo Fisher), passed through a 40 µm filter-capped test tube (Corning, Corning, NY), and analyzed using either a CytoFLEX S flow cytometer (Beckman Coulter, Brea, CA) or a BD Influx FACS machine (BD Biosciences, Franklin Lakes, NJ) or a BD FACSMelody (BD Biosciences). For each experiment, 10,000 events were recorded. During flow cytometry analysis, we gated for cells expressing the Cas12a construct (mCherry$^+$), but not the CRISPR array construct. That is because array processing by Cas12a severs the upstream reporter gene from the poly-A tail, thus potentially disturbing reporter gene expression and thereby the analysis. For *Figures 1 and 3*, three replicates were performed for each sample.

## RT-qPCR to quantify endogenous gene activation (Figure 5)

Cells were transfected as described above. For cell harvesting, all cells in each well were dissociated and included in the analysis and were thus not sorted based on uptake of Cas12a or CRISPR array plasmids. Two biological replicates were performed. Total RNA was extracted with the RNeasy Plus Mini Kit (Qiagen, Germany), according to manufacturer's instructions. Reverse transcription was performed using iScript cDNA Synthesis kit (Bio-Rad, Hercules, CA). Quantitative PCR reactions were run on a LightCycler thermal cycler (Bio-Rad) with iTaq Universal SYBR Green Supermix (Bio-Rad). ΔΔCt values for the target genes were divided by those of *RPL13A* to obtain relative expression. gRNA spacers and RT-qPCR primers are listed in *Supplementary file 3*.

## Assembly of CRISPR arrays

CRISPR arrays were assembled using an oligonucleotide duplexing and ligation method that we developed. First, arrays were designed computationally using SnapGene software (v. 5.1–5.2; Insightful Science, San Diego, CA). The arrays were designed to include two flanking sequences containing a

20 bp overlap with the opened backbone plasmid, as required for a subsequent In-Fusion reaction. This double-stranded CRISPR array sequence was then computationally divided into ≤60 nt DNA sequences with unique 4-nt 5' overhangs, which were ordered from Integrated DNA Technologies (IDT, Coralville, IA) in LabReady formulation (100 μM in IDTE buffer, pH 8.0) and standard desalting purification. For each ligation vial, an *oligonucleotide mix* was first made containing 1 μl of each oligonucleotide. Up to 16 single-stranded oligonucleotides (i.e. corresponding to eight oligo duplexes) were ligated per reaction vial. For CRISPR arrays longer than that, the reaction was divided into multiple vials, each vial containing ≤8 oligonucleotide duplexes (e.g. if the array consists of 12 oligonucleotide duplexes, the reaction was performed in two vials with six duplexes in each).

## Phosphorylation and duplexing

| Oligonucleotide mix | 1.0 μl |
|---|---|
| 2× T7 ligation buffer (NEB) | 2.5 μl |
| H2O | 1.25 μl |
| T4 PNK (NEB) | 0.25 μl |
| Total | 5 μl |

Then run a phosphorylation-duplexing reaction on a thermocycler:

| 37 °C | 30 min |
|---|---|
| 95 °C | 5 min |
| 25 °C | Step down 0.1 °C/s |
| 25 °C | Forever |

Then, add 1 reaction volume (5 μl) of 1× T7 buffer (2.5 μl 2x T7 buffer + 2.5 μl water). Add 1 μl T7 DNA ligase (New England Biolabs, MA) (Important: Use T7 ligase rather than T4 ligase, as T7 ligase lacks the ability to ligate blunt ends). Incubate at 25°C for 3 hr. Then, dilute the sample 1/5 by adding 40 μl water. Run the sample on a 2% agarose gel. A ladder pattern should be visible. Excise the band corresponding to the ligated product. Depending on whether the entire CRISPR array was assembled in a single vial or divided into several vials, do either of the following:

## If the entire array was assembled in a single vial
Gel-purify the excised band using the Macherey-Nagel NucleoSpin Gel & PCR Clean-up kit (Takara Bio, Japan). Insert the purified array into the opened plasmid backbone using In-Fusion cloning (Takara Bio).

## If the array was divided into >1 vial
For all excised bands belonging to the same array, pool the excised bands into a single vial. Gel-purify the pooled bands using the Macherey-Nagel NucleoSpin Gel & PCR cleanup kit. Elute in 15 μl water. Then, add 1 volume (15 μl) of 2× T7 buffer and 1 μl T7 DNA ligase. Incubate at 25 °C for 3 hr. Then, run the ligated product on a 2 % agarose gel. A faint band should be seen corresponding to the full-length CRISPR array. Excise and gel-purify this band. Insert into backbone vector using In-Fusion.

## Design of short CRISPR arrays (two gRNAs) for testing effect of GC content of dummy spacer (Figure 1)
The 51 dummy spacer sequences (*Figure 1*) were adapted from a negative-control sgRNA library generated by *Gilbert et al., 2014*. These sequences correspond to scrambled Cas9 spacer sequences, and we adjusted them slightly for length (20 nt) and GC content. We designed five dummy spacers for each 10%-GC increment (10–90%) because we hypothesized that this amount would sufficiently capture the variability of array performance. We additionally included six dummy spacers from a previous pilot experiment (30%–80% GC). All dummy spacers are listed in *Figure 1—source data 1*.**xlsx**.

## Computation of GC content in sliding window (Figure 1H,J and Figure 2B)

For each of the dummy spacer sequences, we computed the GC content in a sliding 5-nt window (first nucleotides 1–5, then nucleotides 2–6, etc.). For each such window, we then calculated the average and standard error of all 51 spacers. As the sliding window approached the 3' end of the spacers, we reduced the size of the sliding window to 4, then 3, then two nucleotides, in order to increase resolution at the very 3' end. This was performed also for naturally occurring spacers (*Figure 2B*) and CRISPR separators (*Figure 2B*). The spacers we analyzed varied in length from 25 to 36 nt. For this analysis, we truncated the 5' ends of spacers longer than 25 nt. This way, we could align and analyze the 25 nucleotides at the most 3' end of every spacer, even though it meant that we would lose information at the 5' end of longer spacers. For the separator sequences, we first aligned them using the T-Coffee alignment tool (see below), which did not truncate any of the separator sequences.

## Calculation of the predictive power of spacer GC content

For calculating the predictive power of knowing the GC content of three bases in the dummy spacer (*Figure 1K*), we divided each 20-nt spacer into eighteen 3-nt windows and calculated the GC content for each window. For each such window (e.g. nucleotides 1–3), we plotted GC content versus *percent GFP+ cells* for all 51 arrays. We then performed a linear regression (GraphPad Prism v. 9.0; GraphPad Software, San Diego, CA) and used the resulting $R^2$ value for *Figure 1J*.

## Multiple-sequence alignment of naturally occurring CRISPR sequences (*Figure 2*, *Figure 2—figure supplement 1*)

To find bacterial CRISPR-Cas12a operons, we used CRISPR-Cas++ (*Couvin et al., 2018*) using two search strategies: (1) Using the CRISPRCasdb-Blast tool with default settings (accessed September 2020), we input the Cas12a repeat sequences from *Zetsche et al., 2015* and extracted all spacers and repeats, making sure that the spacer sequences were all directed in the 5'-to-3' direction. (2) Using the CRISPRCasdb function, we searched for Cas12a loci in all organisms using default settings. Alignment of separator sequences and post-processed repeats was performed using the multiple-sequence alignment tools of SnapGene (v. 5.1–5.2). The separator sequences were aligned using the T-Coffee algorithm. The sequences used can be found in *Supplementary file 2*.

## Cas12a cleavage assay (Figure 4)

We first PCR-amplified the CRISPR array from our expression plasmids using the KAPA HiFi HotStart polymerase and the KAPA HiFi Fidelity Buffer (KAPA Biosystems, MA) (25 PCR cycles), using primers spanning the array. The forward primer contained the T7 promoter sequence, followed by two Gs for increased expression (TAATACGACTCACTATAGGG) immediately upstream of the primer's target-binding region. The PCR product was purified using the Macherey-Nagel NucleoSpin Gel & PCR Clean-up kit (Takara Bio), and 500 ng DNA was used for the subsequent in vitro transcription reaction. For this, we used the HiScribe T7 Quick High Yield RNA Synthesis Kit (New England Biolabs) according to the manufacturer's protocol and incubated the reaction at 37 °C overnight. RNA was purified using the MEGAclear Transcription Clean-up Kit (Thermo Fisher) using the manufacturer's protocol. For the Cas12a cleavage reaction, we used EnGen Lba Cas12a protein (New England Biolabs) but used our own cleavage buffer rather than NEBuffer 2.1, as we found that this buffer gave better results. This cleavage buffer consisted of Tris-HCl (final concentration 20 mM, pH 7.5, ThermoFisher), KCl (final concentration 50 mM, ThermoFisher), Recombinant RNase inhibitor (final concentration 1 U/µl, Takara Bio), and MgCl$_2$ (final concentration 2 mM). Mg$^{2+}$ is important for the proper folding of the gRNA pseudoknot (*Dong et al., 2016*; *Fonfara et al., 2016*). The concentration of free Mg$^{2+}$ inside living cells is on the order of 1 mM (*Maeshima et al., 2018*). We found that a MgCl$_2$ concentration of 2 mM gave slightly crisper cleavage products than 1, 5, or 10 mM, and we used 2 mM for all experiments.

| Component | Volume |
|---|---|
| Cleavage buffer | 4.3 µl |
| RNA (1 µg/µl) | 0.25 µl |

*Continued on next page*

*Continued*

| Component | Volume |
|---|---|
| Cas12a (100 µM) | 0.45 µl |
| Total | 5 µl |

First, we added 0.25 µl RNA (1 µg/µl stock) to 4.3 µl cleavage buffer, heat-denatured this solution at 70 °C for 2 min and transferred the vials to room temperature. Then we added 0.45 µl Cas12a (100 µM stock solution) and immediately transferred the tubes to a pre-heated thermocycler set at 37 °C. Because each CRISPR array contains three cleavage sites and Cas12a remains bound to its gRNA after processing, we used an excess of Cas12a protein (16:1 Cas12a:array). To stop the reaction, we added 0.3 µl proteinase K (New England Biolabs, 800 U/ml stock solution, which had been pre-diluted 1:1 in cleavage buffer) and transferred vials to a thermocycler pre-set to 50 °C, where they were incubated for 1 hr and then transferred to ice. Samples were analyzed on a 2,100 Bionalyzer (Agilent, CA) using RNA Nano 6000 chips. Samples and Low Range ssRNA Ladder (New England Biolabs) were heat-denatured at 70 °C for 2 min and transferred to ice immediately before loading on the chip in order to minimize secondary structure formation. Bioanalyzer results were normalized based on the sample with the lowest RNA concentration to enable quantitative comparisons between replicates. To calculate how many percent of gRNAs had been fully processed (*Figure 4E*), we used the Bioanalyzer's output of the area of the RNA peak corresponding to fully processed, single gRNA and divided this by the total area of all peaks to estimate the fraction of the total RNA that was contained in the peak corresponding to fully excised, single gRNAs. If processing of these CRISPR arrays proceeds to completion, 28% and 29% of the total RNA mass should be composed of single gRNAs for the two arrays, respectively (e.g., $(41 + 42)/(57 + 41 + 42 + 158) = 0.28$ for the array without the synSeparator; see *Figure 4A*), whereas the remaining RNA mass should consist of the upstream and downstream sequences. We divided the observed fraction of single gRNAs with 0.28 and 0.29, respectively, to find the percentage of maximum processing that had occurred.

## Acknowledgements

We thank Hannah Kempton for vectors encoding Cas12a protein. JPM is supported by the Human Frontier Science Program Long-term Fellowship and the Sweden-America Foundation. LSQ is supported by Li Ka Shing Foundation and the National Institutes of Health Common Fund 4D Nucleome Program (U01 DK127405).

## Additional information

### Competing interests

Jens P Magnusson, Antonio Ray Rios, Lingling Wu, Lei S Qi: The authors have filed a US provisional patent application related to this work (application no. 63/139,095)..

### Funding

| Funder | Grant reference number | Author |
|---|---|---|
| Li Ka Shing Foundation | | Lei S Qi |
| NIH Common Fund 4D Nucleome Program | U01 DK127405 | Lei S Qi |
| Human Frontier Science Program | Long-term Fellowship | Jens P Magnusson |
| Sweden-America Foundation | | Jens P Magnusson |

The funders had no role in study design, data collection and interpretation, or the decision to submit the work for publication.

## Author contributions
Jens P Magnusson, Conceptualization, Formal analysis, Investigation, Methodology, Project administration, Software, Supervision, Visualization, Writing – original draft, Writing – review and editing; Antonio Ray Rios, Investigation, Writing – review and editing; Lingling Wu, Investigation; Lei S Qi, Conceptualization, Funding acquisition, Project administration, Resources, Supervision, Writing – review and editing

## Author ORCIDs
Jens P Magnusson [ID] http://orcid.org/0000-0002-3928-8959
Antonio Ray Rios [ID] http://orcid.org/0000-0002-6717-2267
Lei S Qi [ID] http://orcid.org/0000-0002-3965-3223

## Decision letter and Author response
Decision letter https://doi.org/10.7554/eLife.66406.sa1
Author response https://doi.org/10.7554/eLife.66406.sa2

## Additional files

### Supplementary files
• Supplementary file 1. Spacer, repeat and separator sequences used and analyzed in this study.
• Supplementary file 2. Naturally occurring CRISPR-Cas sequences analyzed in this study.
• Supplementary file 3. Spacers and qPCR primers used for endogenous gene activation (related to *Figure 5*).
• Supplementary file 4. CAGp-FireflyLuciferase-Array-Terminator (*Figure 3C*).
• Transparent reporting form

### Data availability
All data generated or analyzed during this study are included in the manuscript and supporting files. Source data have been provided for all figures.

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
