## [Decision Letter]

**Acceptance summary:**

The Cas12a protein from type V CRISPR-Cas systems can be used for genome editing or modulating gene expression in mammalian cells. One of the advantages of Cas12a over other CRISPR-based systems is the ability to multiplex guide RNAs in a single array. Here, the authors show that processing of RNAs from these arrays can be inhibited by RNA secondary structure, which can be reversed by introducing an artificial sequence, coined the "synSeparator", between guide RNAs. The use of synSeparators increases the activity of Cas12a in mammalian cells, and thus represents an important advance in the development of Cas12a for biotechnology applications.

**Decision letter after peer review:**

Thank you for submitting your article "Enhanced Cas12a multi-gene regulation using a CRISPR array separator" for consideration by *eLife*. Your article has been reviewed by 3 peer reviewers, including Joseph T Wade as the Reviewing Editor and Reviewer #2, and the evaluation has been overseen by Detlef Weigel as the Senior Editor. The following individual involved in review of your submission has agreed to reveal their identity: Chase Beisel (Reviewer #1).

Essential revisions:

1. The authors should test more specific models for how the A/T-rich synSeparators promote crRNA activity, either by analyzing existing data with different separator variants, or experimentally testing more separator/spacer combinations to address specific models. The reviewers recommend focusing on secondary structure in the repeat sequences as a likely source of synSeparator activity.

2. To show that synSeparators are broadly useful for multiplexed Cas12a applications, the authors should test additional targeting crRNA guides, ideally with the design informed by mechanistic insight from Essential Revision #1. The authors should also test crRNAs with synSeparators in another cell line and/or with a different Cas12a protein to further show generalizability.

*Reviewer #1 (Recommendations for the authors):*

Some additional literature should be integrated into manuscript. Fonfara Nature 2016 showed that Cpf1 encodes a distinct domain responsible for processing the transcribed CRISPR array into individual crRNAs. Liao Nat Commun 2019 showed that the order of the spacer in a CRISPR-Cas12a array can immensely impact multiplexed targeting, where targeting by one crRNA was strongly inhibited due to its repeat pairing with the upstream spacer. McCarty Nat Commun 2020 provided a comprehensive review of multiplexing approaches with CRISPR, including the use of CRISPR arrays.

To support general claims, the authors would need to show that the same trends persist when testing different guide sequences. This would include the selected synthetic separators as well as the natural separator.

The authors also need to integrate the folding of the repeat hairpin, as improper folding likely explains the authors' results more-so than general folding of the upstream guide sequence.

References on line 40 should be replaced with the original demonstration of crRNA processing (Brouns Science 2008) and possibly include the original paper characterizing Cas12a (Zetsche Cell 2015).

Descriptions around processing should consider the fact that the spacer is also trimmed to the 20 – 24 nt guide naturally observed with Cas12a crRNAs (Zetsche Cell 2015).

L. 63 – 64: A more reasonable reason for removal of the separator in the original work was that it was seen as dispensable rather than interfering with targeting activity.

Specify somewhere early in the Results which Cas12a was used (e.g. As, Lb, Fn).

L. 301: The claim of a significant increase needs to be supported statistically. Given the smaller fold-changes and the duplicate measurements, many of these increases may not be statistically significant.

*Reviewer #2 (Recommendations for the authors):*

1. The data in Figure 1 correlate predicted secondary structure with crRNA activity. However, this relationship can be tested more directly. I would like to see targeted substitutions in the separator that are predicted to lead to specific changes in secondary structure at key positions around the Cas12a cleavage site.

2. The authors speculate that the separator sequences affect Cas12a processing of the crRNAs, but they rely on indirect readouts of crRNA processing. It would be informative to measure processed crRNA levels directly, especially for the experiment shown in Figure 4. crRNA processing could also be assayed in vitro using purified Cas12a.

3. The data in Figure 4 show a modest effect of introducing a short, A/T-rich separator in the CRISPR array. I think even a small improvement in crRNA activity would be an important advance, so the magnitude of the effect is not a concern. However, I would like to see more than one CRISPR array tested, and it would be informative to individually replace A/T-rich separators with G/C-rich separators. Also, does the spacer order matter? Spacer order could also affect RNA secondary structure.

*Reviewer #3 (Recommendations for the authors):*

1. This is not needed for the current manuscript, but in future experimentation the authors could consider performing comparative small RNA-seq to validate crRNA array processing in presence/absence of the synSeparator.

2. In Figure 4B, the black arrows could be confusing to readers. Consider showing only the green values or more explicitly stating what the black arrows denote.

3. In line 33-36, The word "coding sequence" is not appropriate. The word should be replaced with regulatory element or something similar.

4."permissive" (line 153) should be changed to "permissive".

5. Although authors have mentioned both VPR and mini-VPR (line 530), none of Results section/figures have included VPR. Was full-length VPR used?

6. In Line 202 – legend of Figure 2 there are some extra brackets around "Figure S2A".

[Editors' note: further revisions were suggested prior to acceptance, as described below.]

Thank you for resubmitting your work entitled "Enhanced Cas12a multi-gene regulation using a CRISPR array separator" for further consideration by *eLife*. Your revised article has been evaluated by Detlef Weigel (Senior Editor) and a Reviewing Editor.

The Reviewing Editor has read through your revised manuscript and response letter. We appreciate the impressive effort you have made to respond to the comments from the previous round of review. In principle, the paper is now suitable for publication in *eLife*, pending a response to a few small comments ; these comments should all be addressable with changes to the text. The most important issue relates to Figure 4. The new in vitro assay is a nice addition since it suggests the effects of RNA secondary structure are directly on Cas12a. However, I am a little confused by the data presentation in Figure 4, and I think you may be overstating the magnitude of the difference in cleavage between RNAs with/without a SynSeparator. There is clearly an improvement from adding the SynSeparator, which alone is sufficient reason to include these new data. However, you should either more clearly explain/present the data, or soften your conclusions. See comment #3 below for more details.

Specific comments:

1. Lines 578-585 (Discussion). This paragraph doesn't relate to anything in the current study, so I recommend removing it.

2. The Discussion covers a lot of ground. I recommend adding section titles to improve the readability.

3. Figure 4E summarizes the data for the in vitro cleavage assay. However, to my eye at least, the data in panels B and C, and the data in the associated supplementary figure, don't match panel E. Specifically, without the SynSeparator, the largest increase in cleavage (shown in Figure 4E) occurs between 30 and 60 minutes. However, in panel B, there is almost no difference in the abundance of the 41/42 nt products between 30 and 60 minutes. Moreover, the kinetics of cleavage for the RNA without a SynSeparator are clearly delayed relative to the RNA with a SynSeparator, but cleavage appears to saturate after ~30 minutes (see also panels E and F of the supplementary figure, which to my eye are almost identical). Figure 4E suggests that only 20% of possible cleavage has occurred after 60 minutes for the RNA without a SynSeparator. If that is the case, why is it that cleavage appears to have saturated after 30 minutes? The authors need to present and describe these data more clearly, and potentially soften their conclusions.

4. The authors should use more precise language when describing changes in predicted secondary structure, making clear that these are predictions. For example, rather than saying "as these secondary structures grew tighter, CRISPRa performance gradually worsened", I suggest "as the predicted extent of base-pairing increased, CRISPRa performance gradually worsened".

5. Line 42. This is the one place where I would stick with "crRNA" rather than "gRNA". Perhaps you could mention in parentheses that the RNAs are commonly called crRNAs when expressed from their native loci.

6. Line 42. I suggest "prokaryotic" rather than "bacterial".

---

## [Author Response]

Essential revisions:1. The authors should test more specific models for how the A/T-rich synSeparators promote crRNA activity, either by analyzing existing data with different separator variants, or experimentally testing more separator/spacer combinations to address specific models. The reviewers recommend focusing on secondary structure in the repeat sequences as a likely source of synSeparator activity.

We have now performed several new experiments and computational analyses to explore the mechanism by which synSeparators enhance CRISPR array performance (These experiments are also described in response to specific reviewers’ comments). After performing these experiments, we are confident that appending A/T-rich synSeparators does enhance the efficiency of crRNA array cleavage, which explains the better multiple gene activation we observed in cells. These efforts include:

First, we performed an in vitro Cas12a cleavage assay where we investigated Cas12a’s ability to process CRISPR arrays containing spacers with high or low GC content, and with or without the synSeparator. We now present these data in a new main figure (Figure 4) and an associated supplemental figure (Figure S4). We found that array processing is less efficient when the upstream dummy spacer has high GC content (70%) than when it has low GC content (30%) (Figure 4E). Inserting an AT-rich 4-nt synSeparator between each gRNA rescues this processing defect and makes processing of the array with the 70%-GC spacer nearly as efficient as that with the 30%-GC spacer

(Figure 4E). We present a time-course measurement of cleavage where we subjected CRISPR arrays to Cas12a processing for different amounts of time. Under these experimental conditions, fully processed, single gRNAs are detectable after 10 minutes, at which time the synSeparator-containing array already shows increased processing efficiency compared to the array without the synSeparator (Figure 4B-E). These results are consistent with the functional performance of these very same arrays in cells (Figure 3D). The results demonstrate that high-GC spacers can impair array processing and that an AT-rich synSeparator can rescue poor processing of these arrays. Perhaps surprising is the observation that some processing did occur of the array with the 70%-GC dummy spacer, even without the synSeparator. This is surprising because this same array allowed virtually no GFP activation in cells (Figure 3D). We have added a paragraph in the Discussion where we speculate that the experimental conditions in the cleavage assay may not perfectly mimic those in a living cell. For example, the cleavage assay is in principle allowed to proceed to completion, whereas RNA degradation in a cell might render CRISPR arrays non-functional if they are not processed fast enough.

Second, we have performed a new experiment where we introduced a series of targeted point mutations in the dummy gRNA that were designed to provoke secondary structure formation. The purpose of this experiment was to gain a better mechanistic understanding of where the most disruptive secondary structures are located. Specifically, we asked whether a GFP-targeting gRNA would be most sensitive to secondary structures that form exclusively within the upstream (dummy) spacer, or to structures that also involve the repeat region of the GFP-targeting gRNA itself. We present these results in a new supplemental figure (Figure S3) and an accompanying paragraph in the main text. To address this question, we generated several variants of a short CRISPR array where we gradually mutated the dummy spacer to form secondary structures that were either localized to the dummy spacer or that competed with the natural pseudoknot structure in the GFP-gRNA’s repeat region (Figure S3A, Table S1). These designs were guided by a secondary structure prediction tool, RNAfold. Results showed that, as these secondary structures grew tighter and tighter, CRISPRa performance gradually worsened (Figure S3B). Interestingly, this happened both when the secondary structure was localized to the dummy spacer and when it also involved the GFP-targeting gRNA’s repeat region. This indicated that CRISPR array performance is sensitive to both kinds of disruption. We then included the AAAT synSeparator to these worst-performing arrays and found that this significantly rescued CRISPRa performance in both cases (Figure S3B), suggesting that the separator acts both by insulating gRNAs from each other and by breaking up secondary structures that involve two consecutive gRNAs.

Third, we have performed an experiment to explore the possibility that there is a positional effect in a CRISPR array, such that a gRNA may perform differently depending on where in an array it is located. This experiment could indicate the extent to which global RNA structure affects gRNA function. We present these data in Figure S3C-D and an accompanying paragraph in the main text. We generated four CRISPR arrays, each containing one GFP-targeting gRNA and three non-targeting dummy gRNAs, and placed the GFP-targeting gRNA in each of the four positions. We made sure that the GFP-targeting gRNA would be preceded by the same dummy gRNA in as many arrays as possible, because different dummy gRNAs may have different propensities to generate secondary structures immediately upstream of the GFP-targeting gRNA, which would add variability. We found no positional effect in the performance of the GFP-targeting gRNA. This suggests that local RNA structure may be a more important determinant of gRNA performance than global RNA structure, at least in this experimental setting.

Fourth, we have performed a new computational analysis of spacer secondary structure using RNAfold, which contributes to a mechanistic understanding of CRISPR array processing in the context of high-GC spacers: We have assessed predicted secondary structure formation of the dummy spacers from Figure 1G (presented in the new Figure 1M). These results show that the performance of a GFP-targeting gRNA is anticorrelated with the predicted secondary structure of the upstream dummy spacer (Figure 1M). Interestingly, this anticorrelation was strongest when the structural prediction included both the non-targeting dummy spacer and the subsequent GFP gRNA (R^2^ = 0.57; Figure 1M) rather than only the dummy spacer (R^2^ = 0.27; Figure S1C), suggesting that secondary structures that directly involve the GFP-targeting gRNA may be particularly disruptive. Together with the experimental results described above (with the targeted mutations introduced into the dummy spacer), these results suggest that, even though secondary structures may be disruptive if they are confined to the upstream spacer, they may often be even more disruptive if they also involve the targeting gRNA itself.

Fifth, we present predicted secondary structures (RNAfold) from the two worst-performing CRISPR arrays from Figure 1G in new panels in Figure 3G-H. These structures suggest that the poor performance of these arrays may be caused by tight secondary structures that form between the dummy spacer and the GFP-targeting gRNA, which is likely to impair Cas12a processing (Figure 3H). Interestingly, both the natural L. bacterium separator and the AAAT synSeparator are predicted to break up these structures or form protrusions that may facilitate Cas12a access to its cleavage site. For example, structural ensemble diversity increases from 1.78 to 12.51 for this 50%-GC spacer, and from 7.07 to 12.43 for this 70%-GC spacer, when the full-length natural separator is included.

Taken together, all these new results indicate that the disruptive effect of high-GC spacers is likely caused by secondary structures (Figure 1M) that are either restricted to a single gRNA or involve two consecutive gRNAs (Figures 1M, S1C, S3A-B). CRISPR arrays containing such disruptive gRNAs are processed inefficiently by Cas12a (Figure 4B-E), which likely explains their poor performance. However, AT-rich separator sequences, even short, 4-nt sequences, can break up such secondary structures and give Cas12a access to its cleavage site (Figure 3H), improving CRISPR array processing (Figure 4) and performance (Figure 3D-J).

2. To show that synSeparators are broadly useful for multiplexed Cas12a applications, the authors should test additional targeting crRNA guides, ideally with the design informed by mechanistic insight from Essential Revision #1. The authors should also test crRNAs with synSeparators in another cell line and/or with a different Cas12a protein to further show generalizability.

We have addressed the generalizability of our findings with several new experiments:

First, we now include data demonstrating that a different nuclease-deactivated, enhanced-Cas12a from Acidaminococcus species (denAsCas12a; Kleinstiver et al., 2019) is also sensitive to the effects of high-GC spacers (Figure 3J). This poor performance was largely rescued by including a TTTT synSeparator derived from the natural AsCas12a separator (see Figure 2C). This experiment demonstrates that sensitivity to high-GC spacers is not specific to our particular Lb-dCas12a activator but is a general property of different Cas12a variants, consistent with the observation that all observed Cas12a arrays contain AT-rich separator sequences.

Second, we now include data demonstrating that the beneficial effect of adding a synSeparator is not limited to the AAAT sequence derived from the Lachnospiraceae b. separator. We now include three other 4-nt, AT-rich synSeparators derived from Acidaminococcus s. (TTTT), Moraxella b. (TTTA) and Prevotella d. (ATTT) (Figure 3I). All these synSeparators rescued the poor GFP activation caused by an upstream spacer with high GC content, demonstrating that the capacity to insulate neighboring gRNAs from each other is a general feature of AT-rich sequences. That said, these four synSeparators differed somewhat in the magnitude of GFP rescue they enabled (Figure 3I). This quantitative difference might be due to an intrinsic “insulation capacity” that differs between these sequences, or to the way they interact with the specific Lb-Cas12a protein used in this experiment, or to sequence-specific secondary structures within this particular CRISPR array. We discuss these possibilities in the Discussion.

Third, we attempted to investigate the effect of the synSeparator in different cell types. However, many cell types are difficult to transfect and dCas12a-VPR-mCherry is a big construct (>6 kb). Either due to poor transfection efficiency or poor expression of the Cas12a activator construct, CRISPRa activity was consistently poor in these cell types, both with and without the synSeparator (e.g., we cannot observe fluorescence from the mCherry gene fused to the dCas12a activator, which we always see in HEK293T cells). Because of the low delivery efficiency of CRISPRa, it was not possible to evaluate the performance of the synSeparator. To our knowledge, there have not been many reports using dCas12a-VPR in cell types other than HEK293T. While we think that it will be important to optimize CRISPRa in many cell types (e.g., by optimizing transfection conditions, Cas12a variants, promoters, expression vectors, etc.), the focus of our study has been to show the separator’s mechanism and general function; we believe that optimizing general CRISPRa for different cell types is beyond the scope of this paper. We acknowledge that this is a limitation of our study and we have added a paragraph about this in the Discussion (line 355. All line numbers refer to the document with no markup visible). We nevertheless hypothesize that the negative influence of high-GC spacers and the insulating effect of synSeparators are generalizable across cell types. That is because we could observe improved array processing with the synSeparator even in the cell-free context of an in vitro expression system, as described above (Figure 4). This suggests that the sensitivity to spacer GC content is determined only by the interaction between Cas12a and the array, rather than being dependent on a particular cellular context.

Reviewer #1 (Recommendations for the authors):Some additional literature should be integrated into manuscript. Fonfara Nature 2016 showed that Cpf1 encodes a distinct domain responsible for processing the transcribed CRISPR array into individual crRNAs. Liao Nat Commun 2019 showed that the order of the spacer in a CRISPR-Cas12a array can immensely impact multiplexed targeting, where targeting by one crRNA was strongly inhibited due to its repeat pairing with the upstream spacer. McCarty Nat Commun 2020 provided a comprehensive review of multiplexing approaches with CRISPR, including the use of CRISPR arrays.

We thank the reviewer for these suggestions. We have now included references to Fonfara et al. and McCarty et al. in the text (lines 43, 44). Regarding the Liao et al. reference, we have now performed an experiment exploring the positional effect of gRNAs in an array (Figure S3C-D). We observed no difference in GFP activation when a GFP-targeting gRNA was moved to each position in a 4-gRNA array, indicating that position is not necessarily an important determinant of gRNA performance. This finding is interesting considering the positional effect observed in Liao et al. We speculate that positional effects might be observed in cases where local secondary structures exist in some regions of an array but not in others. In such cases, positioning a gRNA in such a region may lead to decreased performance. We have included this discussion in the Results section (line 288).

To support general claims, the authors would need to show that the same trends persist when testing different guide sequences. This would include the selected synthetic separators as well as the natural separator.

We performed an experiment intended to investigate the effect of the synSeparator in different cell types. However, either due to poor transfection efficiency or poor expression of the Cas12a activator construct, CRISPRa activity was consistently poor in these cell types, both with and without the synSeparator (e.g., we did not visually observe fluorescence from the mCherry gene fused to the dCas12a activator, which we always see in HEK293T cells). Because of the low general delivery efficiency of CRISPRa, it was not possible to evaluate the performance of the synSeparator. Many cell types are difficult to transfect and dCas12a-VPR-mCherry is a big construct (>6 kb). To our knowledge, there have not been many reports using dCas12a-VPR in cell types other than HEK293T. While we think that it will be important to optimize CRISPRa in many cell types (e.g., by optimizing transfection conditions, Cas12a variants, promoters, expression vectors, etc.), the focus of our study has been to show the separator’s mechanism and general function; we believe that optimizing general CRISPRa for different cell types is beyond the scope of this paper. We acknowledge that this is a limitation of our study and we have added a paragraph about this in the Discussion (line 355). We nevertheless hypothesize that the negative influence of high-GC spacers and the insulating effect of synSeparators are generalizable across cell types. That is because we could observe improved array processing with the synSeparator even in the cell-free context of an in vitro expression system, as described above (Figure 4). This suggests that the sensitivity to spacer GC content is determined only by the interaction between Cas12a and the array, rather than being dependent on a particular cellular context.

The authors also need to integrate the folding of the repeat hairpin, as improper folding likely explains the authors' results more-so than general folding of the upstream guide sequence.

We have now performed a computational analysis using RNAfold where we correlated the performance of all dummy spacers with their predicted secondary structure (Figure 1M). As the reviewer suspected, correlation between predicted RNA structure and array performance was indeed higher when the structural prediction included both the dummy spacer and the entire GFP-targeting gRNA (R^2^ = 0.57; Figure 1M) than when it included only the dummy spacer (R^2^ = 0.27; Figure S1C). This higher correlation suggests that secondary structures that involve the GFP-targeting gRNA play a more important role in our experiment than secondary structures that only involve the dummy spacer. These results are described in the Results section and in the Figure 1 legend.

References on line 40 should be replaced with the original demonstration of crRNA processing (Brouns Science 2008) and possibly include the original paper characterizing Cas12a (Zetsche Cell 2015).

This has now been done.

Descriptions around processing should consider the fact that the spacer is also trimmed to the 20 – 24 nt guide naturally observed with Cas12a crRNAs (Zetsche Cell 2015).

Please see our response to comments (2) and (3) above. We are not sure we fully understood these comments; please let us know if we did not address them fully.

L. 63 – 64: A more reasonable reason for removal of the separator in the original work was that it was seen as dispensable rather than interfering with targeting activity.

Thanks for the insight. This has been adjusted: ”For this reason, and because the separator has been seen as dispensable, the separator has been omitted when Cas12a arrays have been experimentally expressed in eukaryotic cells.”

Specify somewhere early in the Results which Cas12a was used (e.g. As, Lb, Fn).

This has now been done, both in the legend of Figure 1 and in the Results section.

L. 301: The claim of a significant increase needs to be supported statistically. Given the smaller fold-changes and the duplicate measurements, many of these increases may not be statistically significant.

Our plan was to address the significance by instead using a different CRISPR array in a different cell line. Please see point (10) for a description of our efforts.

Reviewer #2 (Recommendations for the authors):1. The data in Figure 1 correlate predicted secondary structure with crRNA activity. However, this relationship can be tested more directly. I would like to see targeted substitutions in the separator that are predicted to lead to specific changes in secondary structure at key positions around the Cas12a cleavage site.

We thank the reviewer for this suggestion. We have now performed the suggested experiment. We used our design of a 2-gRNA CRISPR array (Figure 1D) where the second gRNA targets the GFP promoter and the first gRNA contains a dummy spacer. We generated several versions of this array where we iteratively mutated the dummy spacer to either form a hairpin restricted to the dummy spacer, or a hairpin that would compete with the pseudoknot in the GFP-gRNA’s repeat region (Figure S3A-B). We found that each of these hairpins significantly reduced performance of the GFP-targeting gRNA. These results suggest that interfering with the pseudoknot indeed disrupts gRNA performance, but that also hairpins that presumably don’t interfere with the pseudoknot are detrimental – perhaps by sterically hindering Cas12a from accessing its cleavage site. Interestingly, the AAAT synSeparator partly rescues performance of the worst-performing of these constructs. These results are displayed in the new Figure S3 and discussed in the related part of the Results section.

2. The authors speculate that the separator sequences affect Cas12a processing of the crRNAs, but they rely on indirect readouts of crRNA processing. It would be informative to measure processed crRNA levels directly, especially for the experiment shown in Figure 4. crRNA processing could also be assayed in vitro using purified Cas12a.

We have now performed an in vitro cleavage assay to address this important point. Using in vitro transcription, we generated RNA transcripts of one of the worst-performing two-gRNA arrays containing a 70% GC dummy spacer (see Figure 3A, D), either with or without the AAAT synSeparator upstream of each Cas12a cleavage site. We incubated this array with purified Lb-Cas12a protein for different amounts of time (2, 5, 10, 30, 60 minutes) and analyzed the cleavage products on a Bioanalyzer. The results showed that array processing occurred both in the presence and absence of the synSeparator but that it was more efficient with the synSeparator. As Cas12a incubation proceeded for longer times, more and more single gRNAs were excised, but for all processing times more single gRNAs had been processed in the array containing the synSeparator. Though the exact numbers may not correspond to what would be seen inside living cells, these results indicate that the synSeparator improves Cas12a processing efficiency and kinetics of CRISPR arrays. We now present these results in a new Figure (Figure 4) and a supplemental figure S4. We describe these results in the Results section, describe the methodology in the Methods section and discuss the results in the Discussion.

3. The data in Figure 4 show a modest effect of introducing a short, A/T-rich separator in the CRISPR array. I think even a small improvement in crRNA activity would be an important advance, so the magnitude of the effect is not a concern. However, I would like to see more than one CRISPR array tested,

We attempted to investigate the effect of the synSeparator in different cell types. However, either due to poor transfection efficiency or poor expression of the Cas12a activator construct, CRISPRa activity was consistently poor in these cell types, both with and without the synSeparator (e.g., we did not visually observe fluorescence from the mCherry gene fused to the dCas12a activator, which we always see in HEK293T cells). Because of the low general efficiency of CRISPRa, it was not possible to evaluate the performance of the synSeparator. Many cell types are difficult to transfect and dCas12a-VPR-mCherry is a big construct (>6 kb). To our knowledge, there have not been many reports using dCas12a-VPR in cell types other than HEK293T. While we think that it will be important to optimize CRISPRa in many cell types (e.g., by optimizing transfection conditions, Cas12a variants, promoters, expression vectors, etc.), the focus of our study has been to show the separator’s mechanism and general function; we believe that optimizing general CRISPRa for different cell types is beyond the scope of this paper. We acknowledge that this is a limitation of our study and we have added a paragraph about this in the Discussion (line 355). We nevertheless hypothesize that the negative influence of high-GC spacers and the insulating effect of synSeparators are generalizable across cell types. That is because we could observe improved array processing with the synSeparator even in the cell-free context of an in vitro expression system, as described above (Figure 4). This suggests that the sensitivity to spacer GC content is determined only by the interaction between Cas12a and the array, rather than being dependent on a particular cellular context.

and it would be informative to individually replace A/T-rich separators with G/C-rich separators.

This is an interesting suggestion. We think that our data already provide an answer to this question: First, our data using a 2-gRNA array (Figure 1) show that if the last ~4 bases in the first spacer are GC-rich, they negatively influence performance of the downstream gRNA (Figure 1H-K). These GC-rich bases in effect act like GC-rich separators for the purpose of this question. Furthermore, replacing the AT-rich synSeparator with a single G can, in extreme cases, lead to severe malfunction of the subsequent gRNA (Figure 3D). We believe these data demonstrate that Gs and Cs can be detrimental when located just upstream of the Cas12a cleavage site.

Also, does the spacer order matter? Spacer order could also affect RNA secondary structure.

We thank the reviewer for this suggestion. Indeed, a previous study (Liao et al., Nat Comm, 2019) showed that the order of the gRNAs in a CRISPR-Cas12a array can impact gRNA performance. We have now performed an experiment exploring the positional effect of gRNAs in an array (Figure S3C-D). We observed no difference in GFP activation when a GFP-targeting gRNA was moved to each position in a 4-gRNA array, indicating that position is not necessarily an important determinant of gRNA performance. This finding is interesting considering the positional effect observed in Liao et al. We speculate that positional effects might be observed in cases where local secondary structures exist in some regions of an array but not in others. In such cases, positioning a gRNA in such a region may lead to decreased performance. We have included this discussion in the Results section (line 288).

Reviewer #3 (Recommendations for the authors):1. This is not needed for the current manuscript, but in future experimentation the authors could consider performing comparative small RNA-seq to validate crRNA array processing in presence/absence of the synSeparator.

Thanks for the suggestion. We have now performed the suggested experiment using an in vitro cleavage assay to address this important point. Using in vitro transcription, we generated RNA transcripts of one of the worst-performing two-gRNA arrays containing a 70% GC dummy spacer (see Figure 3A, D), either with or without the AAAT synSeparator upstream of each Cas12a cleavage site. We incubated this array with Lb-Cas12a protein for different amounts of time (2, 5, 10, 30, 60 minutes) and analyzed the cleavage products on a Bioanalyzer. The results showed that array processing occurred both in the presence and absence of the synSeparator but that it was more efficient with the synSeparator. As Cas12a incubation proceeded for longer times, more and more single gRNAs were excised, but for all processing times more single gRNAs had been processed in the array containing the synSeparator. Though the exact numbers may not correspond to what would be seen inside living cells, these results indicate that the synSeparator improves Cas12a processing efficiency of CRISPR arrays. We now present these results in a new Figure (Figure 4) and a supplemental figure S4. We describe these results in the Results section, describe the methodology in the Methods section and discuss the results in the Discussion.

2. In Figure 4B, the black arrows could be confusing to readers. Consider showing only the green values or more explicitly stating what the black arrows denote.

We now added a more explicit explanation for the green and black values in the legend of Figure 5.

3. In line 33-36, The word "coding sequence" is not appropriate. The word should be replaced with regulatory element or something similar.

We realize that this sentence was confusing. We meant that recruitment to the coding sequence (gene body) can mediate repression. But because it was written in a confusing way, we now simply state “When this fusion protein is recruited to a target gene […] the target gene can be activated or repressed.”

4."permissive" (line 153) should be changed to "permissive".

Could the reviewer please clarify this point? (now on line 131).

5. Although authors have mentioned both VPR and mini-VPR (line 530), none of Results section/figures have included VPR. Was full-length VPR used?

Full-length VPR was used for activation of endogenous genes (Figure 5). We acknowledge that this was not stated clearly enough so we have now clarified this in the Results section (Line 318) and Methods (Line 632).

6. In Line 202 – legend of Figure 2 there are some extra brackets around "Figure S2A".

We have now removed these extra brackets.

[Editors' note: further revisions were suggested prior to acceptance, as described below.]

Specific comments:1. Lines 578-585 (Discussion). This paragraph doesn't relate to anything in the current study, so I recommend removing it.

Are you referring to the paragraph about the endogenous enzyme responsible for processing the 3’ end of each gRNA? (I think line numbers in Word shift depending on whether the original or edited version is displayed). We have now removed this paragraph.

2. The Discussion covers a lot of ground. I recommend adding section titles to improve the readability.

We have now added subheadings in italics (“Considerations for activation of endogenous genes”, “GFP as readout of CRISPR array performance”, “Suggested mechanism of action for the synSeparator”, “Generalizability and outlook”). Please feel free to edit these subheadings if you think others would be more suitable.

3. Figure 4E summarizes the data for the in vitro cleavage assay. However, to my eye at least, the data in panels B and C, and the data in the associated supplementary figure, don't match panel E. Specifically, without the SynSeparator, the largest increase in cleavage (shown in Figure 4E) occurs between 30 and 60 minutes. However, in panel B, there is almost no difference in the abundance of the 41/42 nt products between 30 and 60 minutes. Moreover, the kinetics of cleavage for the RNA without a SynSeparator are clearly delayed relative to the RNA with a SynSeparator, but cleavage appears to saturate after ~30 minutes (see also panels E and F of the supplementary figure, which to my eye are almost identical). Figure 4E suggests that only 20% of possible cleavage has occurred after 60 minutes for the RNA without a SynSeparator. If that is the case, why is it that cleavage appears to have saturated after 30 minutes? The authors need to present and describe these data more clearly, and potentially soften their conclusions.

Thank you for this insightful and important observation. We think there are two main questions in your comment.

The first is, why does there appear to be a discrepancy between Figures 4B and 4E, such that 4B suggests a saturation after 30 minutes while 4E suggests that the reaction has not saturated after 30 minutes. We think this apparent discrepancy has two explanations. First, the triplicate experiments that make up Figure 4E at each time point show variability. We have now updated Figure 4 (and uploaded it using the submission system) to include each data point in 4E, which highlights this variability better. For the array without the synSeparator, one can now see that the 60-minute time point consists of two data points that show reduced cleavage (only slightly higher than the 30-minute time point) and one data point that shows slightly higher cleavage. While the average value for the 60-minute time point is thus shifted slightly upward, the two lower data points may be more representative. In Figure 4B, each curve represents only one representative sample rather than the average of all three samples, as noted in the Figure 4B legend. Second, cleavage efficiency was calculated as the percentage of RNA contained in the 41/42-nt peak divided by total RNA in all cleavage peaks, as described in the Methods section. This means that the calculation is not only sensitive to an increase in the 41/42-nt peak but also to a decrease in larger cleavage products. It appears to be the case that the samples shown in Figure 4B (which are the same as shown in supplementary Figure 4E and F) show a slight decrease in the ~100-nt cleavage product at the 60-minute time point. This would make the 41/42-nt peak make up a larger fraction of the total RNA, even though the 41/42-nt peak appears not to be greater in size.

The second question is, why does the cleavage reaction appear to slow down or saturate already at 30 minutes, despite the excess of Cas12a molecules relative to RNA molecules? We don’t have an answer to this question but speculate that this could be due to a decrease in available Cas12a molecules with time. Because Cas12a remains bound to its gRNA after processing, the amount of freely available Cas12a molecules decreases as the processing reaction proceeds. Perhaps processing of the recalcitrant high-GC gRNAs requires is so unfavorable that it requires a high excess of Cas12a protein to take place. In this scenario, cleavage would become less and less efficient as the amount of freely available Cas12a molecules decreases. We have added this speculation in the Discussion: “… the reaction appears to slow down after 30 minutes, possibly due to the lower concentration of available Cas12a molecules as more and more Cas12a molecules become sequestered by their RNA cleavage products and unavailable to participate in further processing events.”

We hope that these changes facilitate a clearer interpretation of our data.

4. The authors should use more precise language when describing changes in predicted secondary structure, making clear that these are predictions. For example, rather than saying "as these secondary structures grew tighter, CRISPRa performance gradually worsened", I suggest "as the predicted extent of base-pairing increased, CRISPRa performance gradually worsened".

We have performed the suggested change.

5. Line 42. This is the one place where I would stick with "crRNA" rather than "gRNA". Perhaps you could mention in parentheses that the RNAs are commonly called crRNAs when expressed from their native loci.

This change has been performed and the suggested clarification added. (…also known as CRISPR-RNAs [crRNAs] when expressed from their native loci”).

6. Line 42. I suggest "prokaryotic" rather than "bacterial".

This has now been changed.